# Antidiabetic and cardiovascular beneficial effects of a liver-localized mitochondrial uncoupler

Naohide Kanemoto[1,10], Takashi Okamoto[1,10], Koji Tanabe[2,10], Takahiro Shimada[3], Hitomi Minoshima[4], Yuya Hidoh[3], Masashi Aoyama[5], Takashi Ban[6], Yusuke Kobayashi[1], Hikaru Ando[1], Yuki Inoue[7], Motohiro Itotani[8] & Seiji Sato[9]

Inducing mitochondrial uncoupling (mUncoupling) is an attractive therapeutic strategy for treating metabolic diseases because it leads to calorie-wasting by reducing the efficiency of oxidative phosphorylation (OXPHOS) in mitochondria. Here we report a safe mUncoupler, OPC-163493, which has unique pharmacokinetic characteristics. OPC-163493 shows a good bioavailability upon oral administration and primarily distributed to specific organs: the liver and kidneys, avoiding systemic toxicities. It exhibits insulin-independent antidiabetic effects in multiple animal models of type I and type II diabetes and antisteatotic effects in fatty liver models. These beneficial effects can be explained by the improvement of glucose metabolism and enhancement of energy expenditure by OPC-163493 in the liver. Moreover, OPC-163493 treatment lowered blood pressure, extended survival, and improved renal function in the rat model of stroke/hypertension, possibly by enhancing NO bioavailability in blood vessels and reducing mitochondrial ROS production. OPC-163493 is a liver-localized/targeted mUncoupler that ameliorates various complications of diabetes.

[1] Department of Lead Discovery Research, New Drug Research Division, Otsuka Pharmaceutical Co., Ltd., 463-10 Kagasuno Kawauchi-cho, Tokushima 771-0192, Japan. [2] Department of CNS Research, New Drug Research Division, Otsuka Pharmaceutical Co., Ltd., 463-10 Kagasuno Kawauchi-cho, Tokushima 771-0192, Japan. [3] Department of Drug Metabolism and Pharmacokinetics, Nonclinical Research Center, Tokushima Research Institute, Otsuka Pharmaceutical Co., Ltd., 463-10 Kagasuno Kawauchi-cho, Tokushima 771-0192, Japan. [4] Pharmaceutical Planning Group, Otsuka Pharmaceutical Co., Ltd., 2-16-4 Shinagawa Grand Central Tower Minatominami Minato-ku, Tokyo 108-8242, Japan. [5] Biology and Translational Research Unit, Department of Medical Innovations, New Drug Research Division, Otsuka Pharmaceutical Co., Ltd., 463-10 Kagasuno Kawauchi-cho, Tokushima 771-0192, Japan. [6] Department of Renal and Cardiovascular Research, New Drug Research Division, Otsuka Pharmaceutical Co., Ltd., 463-10 Kagasuno Kawauchi-cho, Tokushima 771-0192, Japan. [7] Department of Drug Safety Research, Nonclinical Research Center, Tokushima Research Institute, Otsuka Pharmaceutical Co., Ltd., 463-10 Kagasuno Kawauchi-cho, Tokushima 771-0192, Japan. [8] Quality Assurance Section (Tokushima Wajiki Factory), Quality Assurance Department, Headquarters for Product Safety and Quality Assurance, Otsuka Pharmaceutical Co., Ltd., 306-2 Otsubo Koniu aza Naka-cho Naka-gun, Tokushima 771-5209, Japan. [9] Medicinal Chemistry Research Laboratories, New Drug Research Division, Otsuka Pharmaceutical Co., Ltd., 463-10 Kagasuno Kawauchi-cho, Tokushima 771-0192, Japan. [10]These authors contributed equally: Naohide Kanemoto, Takashi Okamoto, Koji Tanabe. Correspondence and requests for materials should be addressed to N.K. (email: Kanemoto.Naohide@otsuka.jp)

major component of the pathogenesis of diabetes mellitus (DM) is an imbalance between energy expenditure and calorie intake. Although lifestyle modifications with diet and exercise are the best way to improve early diabetic condition, this in itself is not an adequate solution[1]. As a means of pharmacologically inducing energy expenditure to treat metabolic disorders including DM, mUncoupling has been seen as an attractive mechanism of action (MOA). mUncoupling diminishes mitochondrial membrane potential ($\Delta\psi$) and consequently converts energy derived from the respiratory chain into heat. One of the best-characterized agents that induces mUncoupling is 2,4-dinitrophenol (DNP), which was widely used as a weight-loss drug in the 1930s. However, the FDA banned the use of DNP due to the occurrence of numerous severe adverse effects including fatal hyperthermia, which could be ascribed to systemic mitochondrial uncoupling[2,3]. Nevertheless, interest in mUncoupling as a therapeutic strategy[4,5] and attempts to seek safe chemical uncouplers[6–10] have recently been revived.

The liver is a major organ that regulates blood glucose homeostasis by switching and adjusting hepatic glucose production and utilization. It is well known that hepatic glucose output is pathologically increased in type 2 DM patients[11,12]; therefore, treating DM by targeting hepatic glucose metabolism is highly promising. Such attempts, however, have not yet been completely successful[13]. In addition, systemic toxicity could be avoided by enhancing mUncoupling specifically in the liver because the liver metabolic contribution as a fraction of whole body resting energy expenditure was calculated at just 17–18%[14]. Therefore, treating DM with a liver-specific mUncoupler is a reasonable strategy in terms both of efficacy and of safety, as proposed by Shulman and coworkers[8,10].

Through the serendipitous finding of a lead compound and its optimization, we identify a unique mUncoupler, OPC-163493, which is predominantly distributed to the liver and kidney after oral administration. These unique pharmacological properties lead us to systematically and intensively investigate the MOA, efficacy, safety, and potential for clinical application of OPC-163493. Those investigations elucidate that OPC-163493 is a safe mUncoupler applicable to various DM conditions.

## Results

**mUncoupling activity of OPC-163493.** First, we serendipitously discovered cyanotriazole derivatives that activate the TCA cycle. These compounds accelerated carbon dioxide evolution from CHO cells stably expressing a sodium-dependent citrate transporter (NaCT/SLC13A5)[15,16], and are engineered primarily for screening of NaCT inhibitors. When C14-labeled citrate was added to the CHO cells, the compounds reduced radioactivity inside the cells as if these compounds inhibited [14C]-citrate uptake via NaCT. However, the activity of these compounds for reducing the intracellular radioactive count disappeared in the presence of antimycin A, an mitochondrial respiratory chain (MRC) complex III inhibitor (Supplementary Fig. 1a). In reality, these compounds rapidly converted [14C]-citrate into [14C]-$CO_2$ through TCA cycle activation and volatilized it. Next, the derivatives were elucidated as mUncouplers which caused TCA cycle activation. OPC-163493 is an optimized compound selected based on various pharmacological criteria: antidiabetic efficacy, safety, pharmacokinetic properties, and so on. The chemical structure is shown in Fig. 1a.

A mitochondrial swelling assay[17] showed that OPC-163493 is a protonophore with milder activity than carbonyl cyanide-p-trifluoromethoxyphenylhydrazone (FCCP) (Fig. 1b and Supplementary Fig. 1b).

We examined the effects of OPC-163493 on $\Delta\psi$ and mitochondrial ROS production in isolated rat liver mitochondria.

OPC-163493 significantly reduced $\Delta\psi$ at concentrations from 1.25 μM and this reduction almost reached a plateau at 10 μM. In contrast, significant suppression of ROS production was observed from 0.625 μM and maximum suppression was around 2.5–5 μM (Fig. 1c). As reported previously[18], a subtle decline in $\Delta\psi$ can lead to a sharp decrease in mitochondrial ROS production.

To quantitate mUncoupling activity in intact cells[19], we defined it as the augmentation of oxygen consumption rate ($\Delta$OCR) induced by an uncoupler in the presence of an ATP synthase inhibitor, oligomycin. The addition of oligomycin reduced ATP-turnover respiration from the basal OCR. Addition of OPC-163493 then enhanced $\Delta$OCR due to its uncoupling activity. To confirm whether this $\Delta$OCR was derived from mitochondrial respiration, antimycin A was finally added, completely diminishing $\Delta$OCR to the level of nonmitochondrial respiration (Fig. 1d). Addition of OPC-163493 at various concentrations dose-dependently increased $\Delta$OCR. OPC-163493 significantly increased $\Delta$OCR at a concentration range of 0.313–10 μM (Fig. 1e).

We also measured cellular flux indicators: extracellular acidification rate (ECAR), and carbon dioxide evolution rate (CDER)[20], in addition to OCR. OPC-163493 consistently augmented glycolysis, the TCA cycle, and MRC fluxes in a similar concentration range (Fig. 1f–h). Consequently, it was shown that OPC-163493 accelerates cellular energy expenditure along the flux from glycolysis to the TCA cycle to MRC in HepG2 cells. In the same way, OPC-163493 augmented OCR in rat primary hepatocytes (Supplementary Fig. 1c).

Moreover, we examined the effect of OPC-163493 on fatty acid oxidation (FAO), another flux providing acetyl CoA (AcCoA). When glucose was almost depleted from culture medium (0.5 mM glucose instead of 25 mM) in HepG2 cells, ECAR augmentation by OPC-163493 was diminished. Surprisingly, OPC-163493 augmented OCR during these substrate-limited conditions (Supplementary Fig. 1d), suggesting that the major metabolic flow switched from glycolysis to FAO, utilizing endogenous fatty acids. Etomoxir, an inhibitor of carnitine palmitoyltransferase-1, indeed diminished a large part of the OCR (Supplemental Fig. 1d). A significant increase of FAO-dependent OCR by OPC-163493-consuming endogenous fatty acids was observed from a concentration of 0.313 μM (Fig. 1i). Taken together, these findings show that OPC-163493 can augment cellular energy expenditure and respond flexibly according to the availability of nutrients.

**Pharmacokinetics (PK) studies of OPC-163493.** We next performed a PK study in SD rats after oral administration of OPC-163493 (1 mg kg$^{-1}$, Supplementary Fig. 2a). OPC-163493 was quickly absorbed and the peak plasma concentration reached 0.393 μg mL$^{-1}$ (1.17 μM), 3.5 h after administration. The half-life in these rats was 3.74 h and its bioavailability was 53.5%. The PK data showed that OPC-163493 has a high absorption rate and absolute bioavailability.

A quantitative whole body autoradiography (QWBA) study[21] using male and pregnant female SD rats showed that OPC-163493 was chiefly distributed to the liver and renal cortex. OPC-163493 was not detected in the central nervous system, eye, and femur, and was scarcely detected in muscle and fat. Maternal–fetal transmission was not observed (Supplementary Fig. 2b and Supplementary Tables 1 and 2).

Furthermore, we measured directly the hepatic content after oral administration of [14C]-OPC-163493. The peak plasma and hepatic concentration were 0.5371 μg eq mL$^{-1}$ and 2.746 μg eq g-tissue$^{-1}$, respectively (Supplementary Fig. 2c and Supplementary Table 3). The Kp (liver/plasma) values at 2 and 4 h were 4.9 and

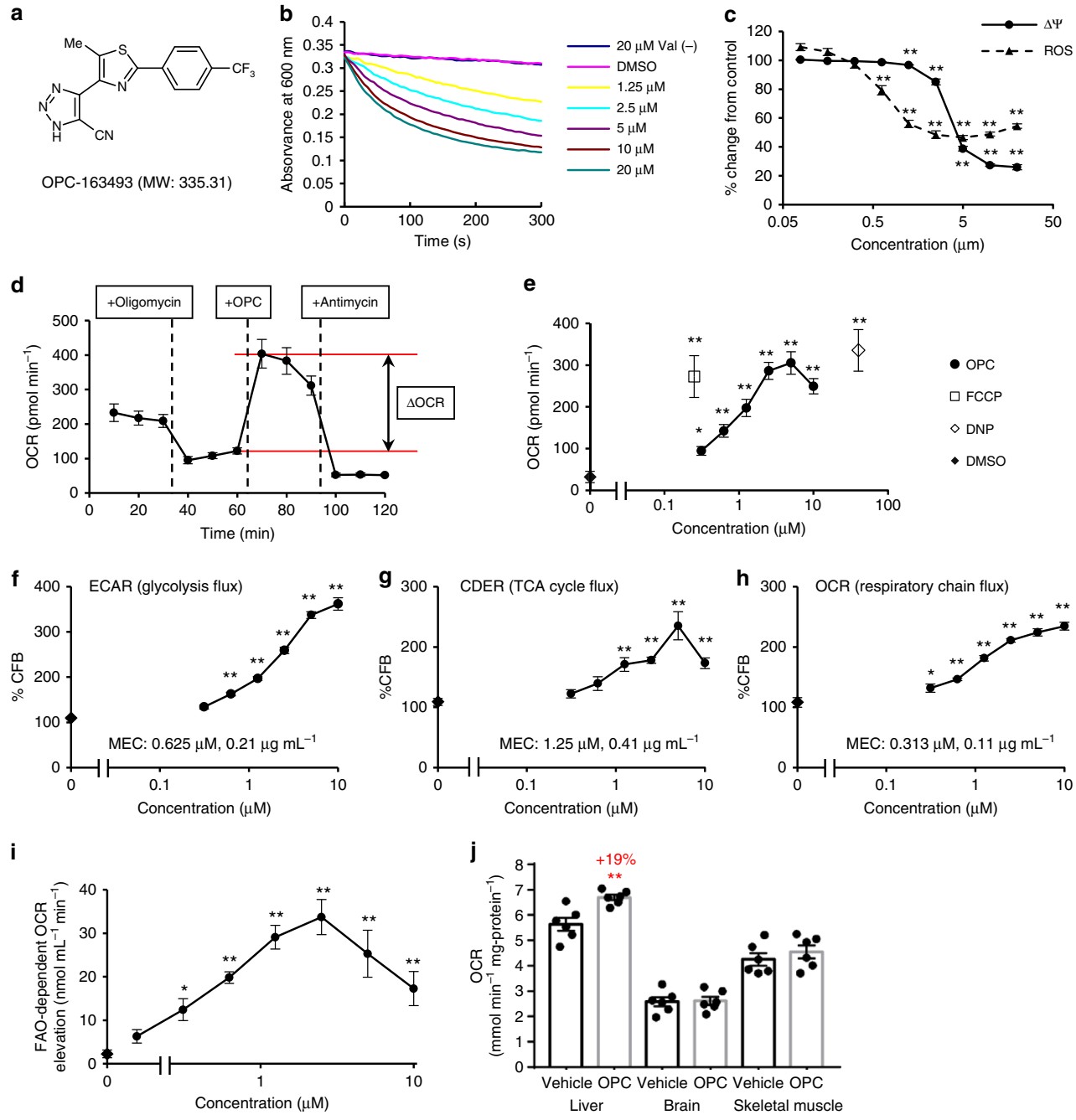

6.6, respectively. Therefore, these data were comparable to those of the QWBA study.

**The effect of OPC-163493 on OCRs in three major organs.** To identify the target organ and the safety profile of OPC-163493, we measured mitochondrial OCRs of tissue homogenates of liver, brain, and skeletal muscle from SD rats orally administered OPC-163493 (10 mg kg$^{-1}$). Although OPC-163493 had no influence on the mitochondrial OCR of both brain and skeletal muscle, its administration significantly augmented OCR in the liver homogenate. The liver OCR of the OPC-163493 treatment group was $6.694 \pm 0.112$ nmol min$^{-1}$ mg-protein$^{-1}$ (Mean ± SE), which was significantly increased by 19% compared with that of the vehicle control group ($5.640 \pm 0.255$ nmol min$^{-1}$ mg-protein$^{-1}$) (Fig. 1j). Two hours after oral administration of OPC-163493, its plasma concentration was $2.917 \pm 0.409$ μg mL$^{-1}$

(Mean ± SE). From these results, OPC-163493 was shown to enhance energy expenditure in the liver, but not in the brain and skeletal muscle, suggesting that OPC-163493 does not induce systemic mUncoupling.

**Antidiabetic effects of OPC-163493 in a wide variety of model animals.** Next, we examined whether OPC-163493 has an antidiabetic effect in multiple animal models. First, to assess the effect in an animal model of type 2 DM, we conducted an oral dosing study in Zucker diabetic fatty (ZDF) rats[22–24]. During the treatment, the HbA1c change from baseline in the vehicle group was 3.4 percentage points, meanwhile those of 1, 2, 4 and 10 mg kg$^{-1}$ day$^{-1}$ OPC-163493-treated groups were 3.3, 3.0, 2.9, and 2.1 percentage points, respectively. OPC-163493 treatment dose-dependently suppressed HbA1c elevation. Significant efficacy was noted at 2 mg kg$^{-1}$ day$^{-1}$ and higher doses of OPC

**Fig. 1** Chemical structure, MOA, and organ specificity of a mUncoupler, OPC-163493. **a** Chemical structure of OPC-163493: 5-{5-methyl-2-[4-(trifluoromethyl)phenyl]thiazol-4-yl}-3H-[1,2,3]triazole-4-carbonitrile. **b** Mitochondrial swelling assay of isolated mitochondria from the liver of a male SD rat. Indicated concentration of OPC-163493 was added to isolated mitochondrial solution in isotonic acetate buffer with or without 3 μM valinomycin (Val) at 0 s and absorbance of the solution was measured at 600 nm. Dark blue line, 20 μM without Val; pink line, DMSO control; yellow line, 1.25 μM with Val; blue line, 2.5 μM with Val; purple line, 5 μM with Val; brown line, 10 μM with Val; green line, 20 μM with Val. **c** The effects of OPC-163493 on mitochondrial membrane potential (Δψ) and mitochondrial ROS production in isolated mitochondria. Δψ and ROS production as $H_2O_2$ emission in the isolated SD rat liver mitochondria were measured using fluorescent probes, Safranin O[55] and Amplex UltraRed[56], respectively. Data represent mean ± SE ($n = 4$). Significant efficacy was found at OPC-163493 concentrations at or above 0.625 μM for ROS production and 1.25 μM or greater for Δψ (**$P < 0.01$, OPC vs. DMSO control using a two-tailed Williams' test). Circles and line, Δψ; triangles and dashed line, ROS. **d** Oxygen consumption rate (OCR) measurement in cultured human liver carcinoma HepG2 cells by an extracellular flux analyzer[64]. ΔOCR evoked by OPC-163493 was defined as its mUncoupling activity. **e** mUncoupling activity of OPC-163493 in HepG2 cells. ΔOCR is plotted against OPC-163493 concentrations. The minimum effective concentration (MEC) was 0.313 μM. Reference compounds, FCCP and DNP, were measured at 0.25 μM and 40 μM, respectively. Data represent mean ± SE ($n = 5$) *$P < 0.05$, **$P < 0.01$, OPC, FCCP, or DNP vs. DMSO-treated group. Two-way ANOVA followed by a two-tailed Dunnett's test was performed for OPC-163493 evaluation, two-way ANOVA was used for FCCP and DNP. Black diamond, DMSO control; black circles and line, OPC-163493; open square, FCCP; open diamond, DNP. **f–h** The effect of OPC-163493 on cellular fluxes in HepG2 cells. Cellular flux indicators for glycolysis flux, extracellular acidification rate (ECAR, **f**), and the TCA cycle flux, carbon dioxide evolution rate (CDER, **g**), as well as OCR (**h**) in the absence of oligomycin were measured. Data are presented as mean ± SE ($n = 5$), *$P < 0.05$, **$P < 0.01$, OPC vs. DMSO-treated group by two-way ANOVA followed by a two-tailed Dunnett's test. **i** FAO-dependent OCR induced by OPC-163493 in HepG2 cells supplemented with glucose-depleted medium. Data represent mean ± SE ($n = 3$). *$P < 0.05$, **$P < 0.01$ OPC vs. DMSO control using a two-tailed William's test. Black circles and line, OPC-163493; diamond, DMSO control. **j** OCR measurement of organ homogenate samples from male SD rats 2 h after oral administration of OPC-163493. 10 mg kg$^{-1}$ of OPC-163493 or vehicle solution was orally administered to male SD rats. Two hours later, three organs were removed and homogenized for measurement of OCR. Data represent mean ± SE (duplicate measurements, $n = 6$). An unpaired $t$ test was used for statistical analysis. **$P < 0.01$, OPC vs. vehicle control

(Fig. 2a and Supplementary Table 4). OPC-163493 also significantly lowered fasting blood glucose levels at doses of 4 and 10 mg kg$^{-1}$ day$^{-1}$ (Supplementary Fig. 3a). No significant effects were observed in plasma insulin levels at the end of treatment (Supplementary Fig. 3b). The Cmax and AUC in the group given 2 mg kg$^{-1}$ day$^{-1}$ were 0.4450 μg mL$^{-1}$ and of 4.540 μg h mL$^{-1}$, respectively (Supplementary Table 5).

Second, to examine the antidiabetic effect in an animal model of insulin-depleted DM, we conducted a dosing study with OPC-163493 mixed chow in Akita mice which developed type-1-DM-like hyperglycemia[24–26]. After treatment, the mean HbA1c value in the control group was 11.0%; meanwhile, those of animal groups fed with chow containing 0.005%, 0.01%, and 0.02% OPC-163493 were 10.9%, 9.9%, and 9.0%, respectively. OPC treatment dose-dependently suppressed HbA1c and significant efficacy was noted at 0.01 and 0.02% OPC-163493 (Fig. 2b and Supplementary Table 6). Diurnal ranges of OPC-163493 plasma concentration in animals fed with 0.01% and 0.02% OPC-163493 mixed chow were within the range of 0.6717–1.647 and 1.840–4.246 μg mL$^{-1}$, respectively (Supplementary Table 7).

Third, to show an antidiabetic effect in an animal model of extreme insulin resistance, we conducted an oral dosing study in aged (27-week-old) ZDF rats that were completely resistant to insulin. The age for commencement of treatment was determined by preliminary insulin tolerance tests (Supplementary Fig. 3g). Since the mean HbA1c value was more than 10% at baseline (Supplementary Table 8), the HbA1c change from baseline in the vehicle group rose few percentage points on days 28 and 43. OPC-163493 treatment dose-dependently lowered the HbA1c from baseline, and significant efficacy was noted at 6 and 10 mg kg$^{-1}$ day$^{-1}$ doses of OPC on day 28 and 10 mg kg$^{-1}$ day$^{-1}$ dose on day 43 (Fig. 2c). OPC-163493 did not affect the levels of plasma insulin on day 43 (Supplementary Fig. 3h). PK parameters are shown in Supplementary Table 9.

Finally, we conducted a long-term study in Otsuka Long-Evans Tokushima Fatty (OLETF) rats whose characteristics are those of late-onset of type 2 DM, with a chronic disease course and a comparatively long lifespan compared with ZDF rats[24,27,28]. OPC treatment showed steady and long-lasting efficacy on HbA1c from a dose of 0.02% OPC-163493 mixed chow (Fig. 2d and Supplementary Table 16) and also reduced both oxidative

stress markers, 8-hydroxy-2′-deoxyguanosine (8-OHdG) and 8-isoprostane[29] (Fig. 2e). The plasma concentrations at 6 AM in each OPC-163493-treated group were measured in this study as a substitute for Cmax (Supplementary Table 10), because the maximum concentrations were generally seen at around 6 AM in the case of mixed chow dosing. There was no influence on food intake and body weight at any treated dose of OPC-163493 in these efficacy studies (Supplementary Fig. 3c–f, i–l).

Taken together, it was shown that OPC-163493 had antidiabetic effects in multiple animal models, and the effect was suggested to be independent of insulin.

**The effects of OPC-163493 on respiratory metabolism in ZDF (M) rats**. To investigate the effects on respiratory metabolism using an indirect calorimeter, we conducted a 4-week dosing study with OPC-163493 mixed chow in ZDF(M) rats from the age of 7 weeks. OPC-163493 treatment showed a significant suppression of HbA1c change from baseline compared with the control group (Fig. 2f). After treatment, the time course of oxygen consumption and carbon dioxide production in the OPC-163493 group were generally comparable with those in the control group (Supplementary Fig. 3o, p). Nor did OPC treatment influence spontaneous locomotor activity (Fig. 2g). Consequently, there was no significant difference in energy expenditure (Fig. 2h) and the respiratory exchange ratio (Fig. 2i) between control and OPC-163493 groups after treatment. OPC-163493 did not affect food intake and body weight (Supplementary Fig. 3m, n). Diurnal ranges of OPC-163493 plasma concentration in mixed chow-fed animals were within the range of 3.938–4.536 μg mL$^{-1}$ (Supplementary Table 11). These results show that OPC-163493 does not affect systemic energy expenditure, further supporting the compound's safety.

**Hepatic lipid-lowering effect of OPC-163493 in fatty liver models**. To examine the efficacy of OPC-163493 in an animal model of severely obese type 2 DM, we conducted a dosing study with OPC-163493-mixed chow in ob/ob mice[24,30,31]. During treatment, 0.02% OPC-163493 mixed chow-fed mice showed significantly greater hyperphagia, which increased 15% above that in control chow-fed mice, though those animals experienced body

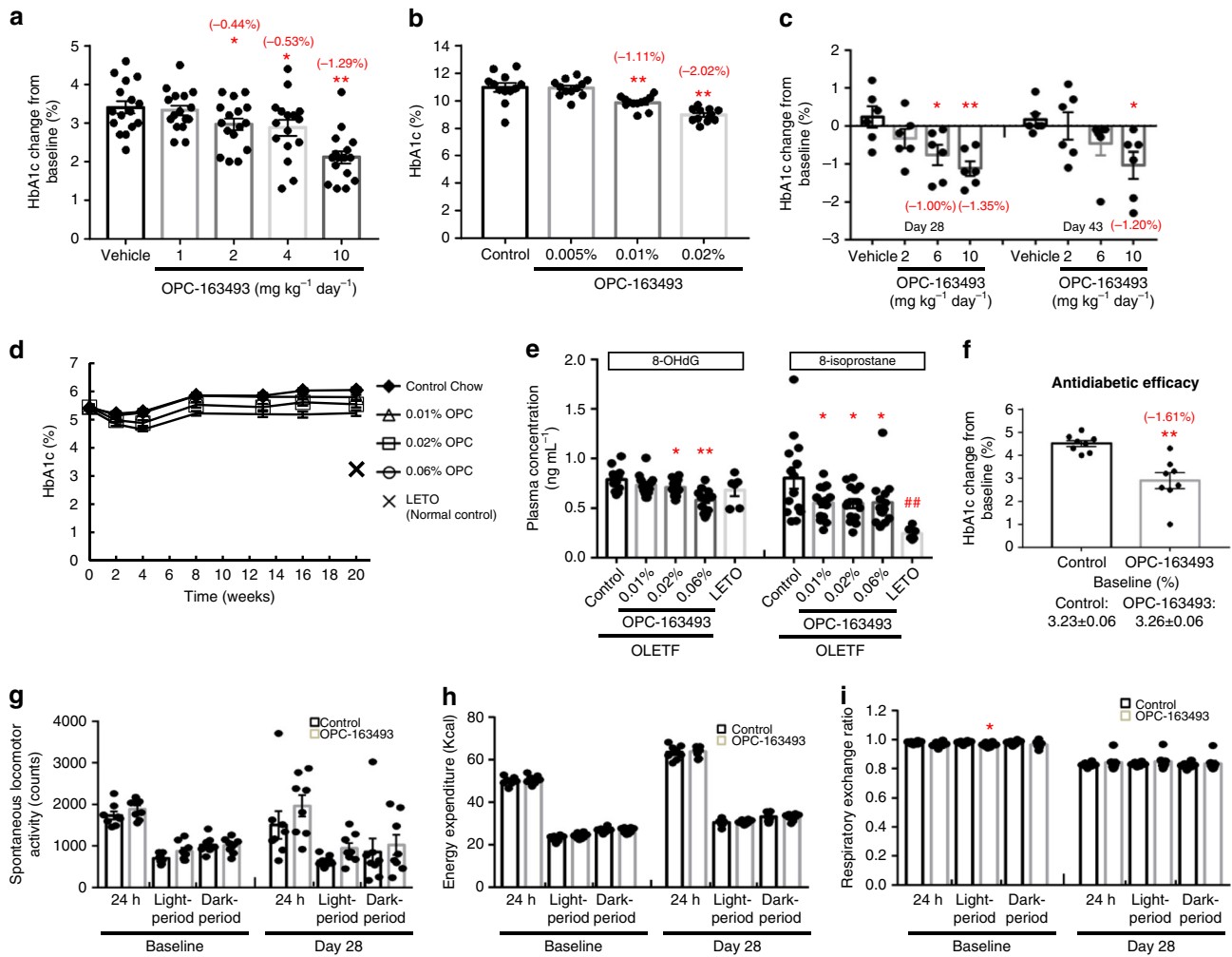

**Fig. 2** Antidiabetic effects of OPC-163493 in a wide variety of animal models. **a** Effect of repetitive oral dosing with OPC-163493 (BID) on HbA1c changes from baseline in male ZDF Rats. OPC treatments were performed on rats aged between 11 and 17 weeks. Data represent mean ± SE ($n = 16$). Significant efficacy was found in the 2, 4, and 10 mg kg$^{-1}$ day$^{-1}$ OPC-163493-treated groups (*$P < 0.05$, **$P < 0.01$, OPC vs. vehicle group using Williams' test with two-way ANOVA). **b** Effect of OPC-163493 dosing with mixed chow on HbA1c values in male Akita mice. OPC treatments were performed on mice aged between 6 and 12 weeks. Data represent mean ± SE ($n = 12$). Significant efficacy was found in the 0.01 and 0.02% OPC-163493-treated groups (**$P < 0.01$, OPC vs. control chow group using a two-tailed Williams' test). **c** Effect of OPC-163493 repetitive oral dosing (BID) on HbA1c changes from baseline in old male ZDF rats having extreme insulin resistance. OPC treatments were performed on rats aged between 27 and 33 weeks. Data represent mean ± SE ($n = 6$). Significant efficacy was found in the 6 and 10 mg kg$^{-1}$ day$^{-1}$ OPC-163493-treated groups on day 28, and in the 10 mg kg$^{-1}$ day$^{-1}$ OPC-163493-treated group on day 43 (*$P < 0.05$, **$P < 0.01$, OPC vs. vehicle group using a two-tailed Williams' test). **d** Long-term effect of OPC-163493 dosing with mixed chow on HbA1c values in male OLETF rats. Twenty-week treatments were carried out between the age of 27 and 47 weeks. LETO (Long-Evans Tokushima Otsuka) rats were used as normoglycemic control rats. Data represent mean ± SE ($n = 14$) except for LETO ($n = 6$). Significant efficacy was found in the 0.02 and 0.06% OPC-mixed chow-treated groups using a mixed model for repeated measures (MMRM) method followed by Dunnett's test (OPC vs. control chow group, $P < 0.01$). Black diamond, control chow; open triangle, 0.01% chow; open square, 0.02% chow; open circle, 0.06% chow; cross, LETO rats with control chow. **e** Long-term effect of OPC-163493 on oxidative stress markers in OLETF rats. Data represent mean ± SE ($n = 14$) except for LETO ($n = 6$). Significant improvements were found in the 0.02 and 0.06% for 8-OHdG, and 0.01, 0.02, and 0.06% OPC-mixed chow-treated groups for 8-isoprostane using a two-tailed Williams' test (OPC vs. control chow group, *$P < 0.05$, **$P < 0.01$). A significant difference was also observed for 8-isoprostane between the LETO rat group and the OLETF rat (control chow) group using an unpaired $t$ test (##$P < 0.01$). **f** Effect of OPC-163493 dosing with mixed chow on HbA1c in male ZDF(M) rats. Each value represents the mean ± SE ($n = 8$). A significant difference was found (**$P < 0.01$, unpaired $t$ test). **g** Effects of OPC-163493 on spontaneous locomotor activity in ZDF(M) rats. Measurements were carried out before (baseline) and after 4 weeks of treatment (day 28). Each value represents the mean ± SE ($n = 8$). No significant difference was found (unpaired $t$ test). Black bar, control; gray bar, OPC-163493. **h** Effects of OPC-163493 on energy expenditure in ZDF(M) rats. Each value represents the mean ± SE ($n = 8$). No significant difference was found (unpaired $t$ test). Black bar, control; gray bar, OPC-163493. **i** Effects of OPC-163493 on respiratory exchange ratio (RER) in ZDF(M) rats. Each value represents the mean ± SE ($n = 8$). A significant difference was found at baseline in the light period (*$P < 0.05$, unpaired $t$ test); however, no significant difference was found after treatment. Black bar, control; gray bar, OPC-163493

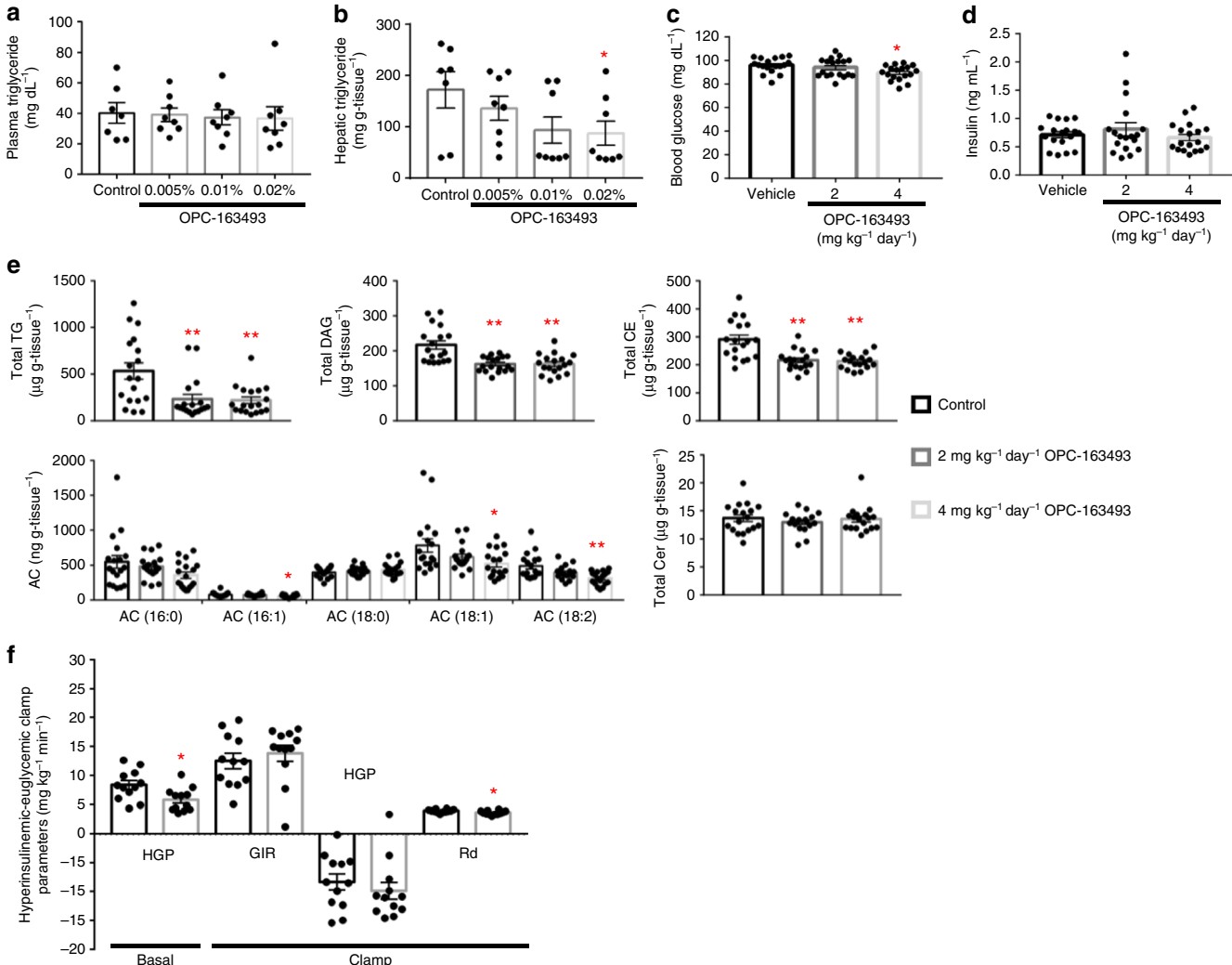

**Fig. 3** Antisteatotic effect and hyperinsulinemic-euglycemic clamp test. **a, b** Effect of OPC-163493 dosing with mixed chow on plasma (**a**) and hepatic (**b**) triglyceride (TG) in male ob/ob mice. Data represent mean ± SE (control group: $n = 7$, other groups: $n = 8$). No significant difference was found in plasma TG using a two-tailed Williams' test. A significant difference was found in hepatic TG between control and 0.02% OPC treatment groups using a two-tailed Williams' test (*$P < 0.05$). Control chow and OPC-163493 admixed chow were fed to each animal group for 10 weeks from the age of 8 weeks. **c, d** Effect of OPC-163493 on fasting blood glucose (**c**) and plasma insulin (**d**) levels in HFD SD Rats. Data represent mean ± SE ($n = 18$). Significant efficacy was found in the 4 mg kg$^{-1}$ day$^{-1}$ OPC-163493-treated groups in blood glucose but not in plasma insulin (*$P < 0.05$ OPC vs. vehicle group by Dunnett's test). **e** Effect of OPC-163493 on hepatic lipids in HFD SD Rats. Data represent mean ± SE ($n = 18$). *$P < 0.05$, **$P < 0.01$ OPC vs. vehicle group by Dunnett's test. Black bar, control; gray bar, 2 mg kg$^{-1}$ day$^{-1}$; silver bar, 4 mg kg$^{-1}$ day$^{-1}$. SD rats were fed with 60 kcal% fat chow for 2 weeks from the age of 10 weeks and the prefeeding was followed by 2-week oral treatment with OPC-163493 during a continued HFD. **f** Hyperinsulinemic-euglycemic clamp test. Male ZDF(M) rats were fed ad libitum with 0.04% OPC-163493 mixed chow or control chow for 3 weeks from the age of 7 weeks. Data represent mean ± SE ($n = 12$). Significant differences between the control and OPC-163493 treatment groups were found in basal HGP and Rd (*$P < 0.05$, unpaired $t$ test). Black bar, control; gray bar, OPC-163493

weight gain similar to that of the control animals (Supplementary Fig. 3q, r). Although the HbA1c-lowering effects in the 0.005% and 0.01% treatment groups were sufficiently effective, hyperphagia in the high dose (0.02%) group statistically hid the HbA1c-lowering efficacy (Supplementary Fig. 3s and Supplementary Table 12). On the other hand, although OPC-163493 treatment did not affect plasma triglyceride (TG) (Fig. 3a), it reduced hepatic TG in a dose-dependent manner. The efficacy (nearly 50% reduction) was statistically significant in the 0.02% treatment group despite overeating (Fig. 3b). The plasma concentrations at 6 AM of OPC-163493 were measured in each group (Supplementary Table 13).

To investigate the antisteatotic effect of OPC-163493 further, we performed lipidomic analysis of hepatic lipids in HFD SD rats,

as a simple animal model of steatosis. During the treatment period, OPC-163493 did not affect body weight and food intake (Supplementary Fig. 3t, u). After 2-week treatment, although there was no difference in plasma insulin levels between the vehicle- and OPC-163493-treated groups, OPC-163493 treatment slightly but significantly reduced fasting blood glucose (Fig. 3c, d). Although OPC-163493 did not alter plasma TG and total cholesterol (Supplementary Fig. 3v), OPC-163493 significantly reduced hepatic total TG, total diacylglycerol (DAG), and total cholesterol ester (CE) by 57%, 25%, and 26%, respectively even in the low-dose (2 mg kg$^{-1}$ day$^{-1}$) OPC-163493-treated group. Long-chain acylcarnitines (AC, C16 and C18) were prominently decreased by OPC-163493 treatment (Fig. 3e and Supplementary Table 14). Overall, sphingomyelins (SM), phosphatidylcholines

(PC), and lysophosphatidylcholines (LPC) including even-numbered long-fatty acids (C16, C18, and C20) were thought to be preferentially consumed (Supplementary Table 14). Ceramides (Cer) were not exceptionally changed by OPC-163493 treatment (Fig. 3e and Supplementary Table 14). We did not find any differences in plasma adipokines between the vehicle and OPC-treated groups (Supplementary Fig. 3w). The PK parameters of OPC-163493 in HFD SD rats are shown in Supplementary Table 15.

**Hyperinsulinemic-euglycemic clamp test of OPC-163493 in ZDF(M) rats.** To elucidate the mechanism of the glucose-lowering action of OPC-163493 in vivo, we conducted a hyperinsulinemic-euglycemic clamp test in ZDF(M) rats. After the treatment, OPC-163493 significantly suppressed blood glucose levels. At the start of the clamp test ($-60$ min; just before glucose infusion began), blood glucose levels of the control and OPC-163493 groups were $338.3 \pm 34.8$ and $219.5 \pm 27.9$ mg dL$^{-1}$, respectively. Blood glucose levels were successfully clamped around the targeted level (140 mg dL$^{-1}$) during the period from 60 to 120 min (Supplementary Fig. 4). OPC-163493 treatment significantly lowered the rate of hepatic glucose production (HGP) at the basal state by 2.52 mg kg$^{-1}$ min$^{-1}$ compared to control treatment. On the other hand, the rate of glucose disappearance (Rd) in the control group was significantly higher than that in the OPC-163493 group, by 0.31 mg kg$^{-1}$ min$^{-1}$ (Fig. 3f and Supplementary Table 17). Since the difference in Rd was markedly less than the differences in HGP and glucose infusion rate (GIR) and those SEs (Supplementary Table 17), the impact of this Rd difference on glucose regulation was very limited or thought to be physiologically meaningless. Although we found a significant difference in HGP between the OPC-163493 and control groups only at the basal state but not at the clamp state, the values at both states were generally lower in the OPC-163493 group (Fig. 3f and Supplementary Table 17). These results suggested that OPC-163493 mainly worked on the liver and reduced the basal HGP but did not improve muscle insulin resistance in ZDF(M) rats.

**Metabolomic analysis of in vitro effects of OPC-163493.** To further investigate the mechanism of the antidiabetic effects of OPC-163493, we conducted a metabolomic analysis to measure changes in metabolic fluxes in HepG2 cells after OPC-163493 treatment. In this CE-QqQ/TOFMS analysis[32–34], 104 of 116 target metabolites were detected (Supplementary Table 18).

At first, regarding the adenylate pool, though an mUncoupler generally competes against ATP synthesis in MRC, ATP was not reduced and total adenylate nucleotides did not change even at the highest dose of OPC-163493. Meanwhile, ADP and AMP were dose-dependently increased. Consequently, ratios of AMP/ATP rose (Fig. 4a). We confirmed the expected induction of phosphorylation of both AMP-activated protein kinase[35–37] (AMPK activation) and its downstream enzyme, acetyl CoA carboxylase[36] (ACC deactivation) in HepG2 cells with OPC-163493 treatment (Fig. 4b).

Although ATP levels did not change, OPC-163493 treatment decreased GTP and phosphocreatine with an increase in GDP, GMP, and creatine. The priority of ATP over GTP and phosphocreatine would lead the high-energy phosphate pool to replenish itself with ATP (Fig. 4c).

Succinate elevation along with fumarate decline was observed in the OPC-163493-treated cells (Fig. 5a, b), suggesting that succinate dehydrogenase (SDH/Complex II) was forced into a resting state despite the circumstance in which both CDER and OCR are augmented (Fig. 1g, h). Considering the presence of

mUncoupler/protonophore, MRC would be driven to maintain $\Delta\psi$ as much as possible; the MRC Complexes I, III, and IV, which are able to pump protons out from the matrix to the intermembrane space, would then take precedence over Complex II, which is not capable of proton pumping. For that reason, electron flow from Complex I, via III, to IV in particular would be strengthened. Supporting this idea, the essential substrate, NADH, was significantly decreased by OPC-163493 treatment from the lowest dose (Fig. 5a).

The stalled SDH would cause accumulation of upstream substrates, succinate and 2-oxoglutarate (2-OG), and a shortage of the downstream substrates fumarate and malate in the TCA cycle (Fig. 5b). In this context the TCA cycle is supposed to mobilize all utilizable anaplerotic fluxes[38–41] in addition to glycolysis flux, to produce as much NADH as possible. The utilization of multiple anaplerotic fluxes was indicated by decreases in several amino acids, alanine, aspartate, proline, and serine in OPC-163493-treated cells (Fig. 5b).

In the glycolytic pathway, the downstream intermediates of the rate-limiting enzymes phosphofructokinase (PFK), fructose 1,6-bisphosphate (F1,6P), glyceraldehyde 3-phosphate and dihydroxyacetone phosphate (DHAP) were moderately reduced and simultaneously 2,3-bisphosphoglycerate (Diphosphoglycerate) and fructose 1-phosphate (D-F1P) were also decreased after OPC-163493 treatment (Fig. 6a). These changes apparently indicated activation of glycolysis. Only glycerol 3-phosphate was increased (Fig. 6a). One possible cause of this increase is stalling of an inner membrane-bound mitochondrial glycerol-3-phosphate dehydrogenase (mGPD) by OPC-163493 treatment. mGPD is also a FAD-dependent enzyme, and reduces Coenzyme Q in the same way as Complex II[42].

OPC-163493 treatment decreased several intermediates in the pentose phosphate pathway (Fig. 6b). This indicated decreased substrate flux into the pentose phosphate pathway and increased efflux to the glycolytic pathway. The increases in ADP and GDP were considered to inhibit phosphoribosylpyrophosphate synthetase (PRPS)[43], which converts ribose 5-phosphate (R5P) into phosphoribosylpyrophosphate (PRPP) (Fig. 6b).

In the purine nucleotide cycle, activation of an anaplerotic flux to provide fumarate by OPC-163493 treatment may be a cause for increases in AMP and IMP[38] (Figs. 5b and 6c).

In the urea cycle, elevation of N-acetylglutamate (N-AcGlu), the obligate activator for carbamoyl phosphate synthetase-1 (CPS1)[41], was observed in the OPC-163493-treated cells. Consequently, conversion of ornithine to citrulline would be accelerated, and subsequent reactions by both argininosuccinate synthetase (AS) and argininosuccinate lyase (AL) seemed to be activated as well. As a result, aspartate and argininosuccinate were decreased while arginine was increased (Fig. 6d). This process is identical to the anaplerotic flux that produces fumarate as shown in Fig. 5b[41].

Another remarkable feature was that the ratio of glutathione to its disulfide form (GSH/GSSG) was increased, indicating the oxidative stress reduction[44], and 3-hydroxy-3-methylglutaryl-CoA (HMG CoA) accumulation induced by OPC-163493 treatment would occur by inhibition of HMG CoA reductase (HMGR) due to AMPK activation[36] (Fig. 6e).

**Metabolomic analysis of long-term and in vivo effects of OPC-163493.** We also conducted another metabolomic analysis as an in vivo study comparing the liver metabolites between baseline controls, 6 weeks of OPC-163493 treatment, and vehicle-treated ZDF rats (Supplementary Fig. 5a, b, and Supplementary Table 19). Regarding the antidiabetic effects of OPC-163493: similar efficacy was again obtained, compared with the previous study (Fig. 2a and Supplementary Fig. 3a). Comparing baseline

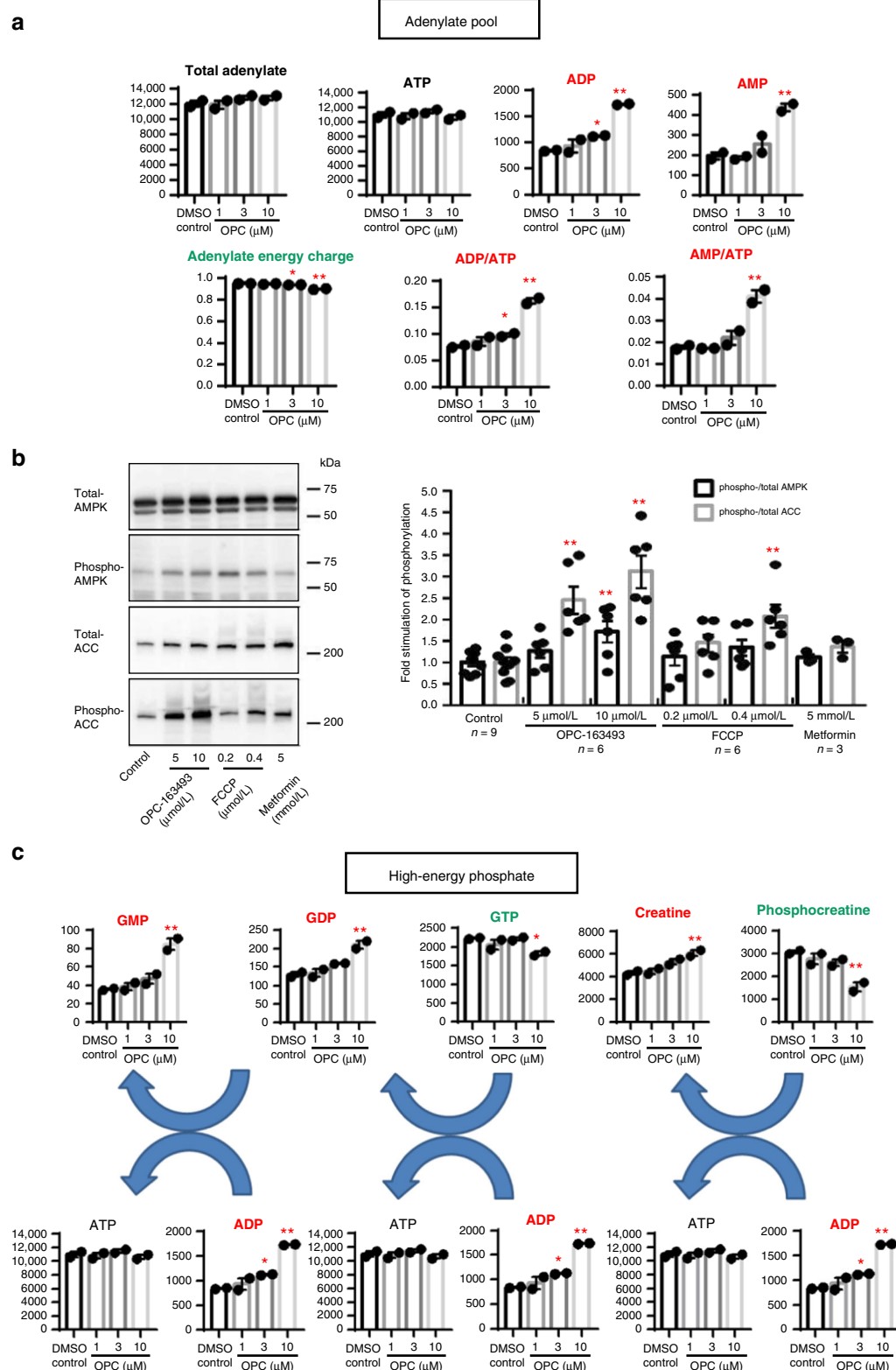

**Fig. 4** Metabolomic analysis of in vitro effects of OPC-163493 in HepG2 cells. Data represent mean ± SE, $n = 2$ for metabolomic analysis (**a** and **c**), *$P < 0.05$, **$P < 0.01$, OPC vs. DMSO control using a two-tailed Williams' test. All units for vertical lines in metabolite graphs are pmol $10^6$ cells$^{-1}$ (except ratios of substrates). When significant increases and decreases were observed, those graph titles were labeled with red and green, respectively. **a** Effects on adenylate pool. Total adenylate = [ATP] + [ADP] + [AMP], Adenylate energy charge = ([ATP] + 0.5 × [ADP])/[Total adenylate]. **b** Phosphorylation of AMPK and ACC in OPC-163493-treated HepG2 cells. Representative data are shown in western blots. Data represent mean ± SE. The numbers in each sample are described in the graph. *$P < 0.05$, **$P < 0.01$, OPC vs. DMSO control using a two-tailed Williams' test. Black bar, pAMPK/total AMPK; gray bar, pACC/total ACC. **c** Effects on high-energy phosphate

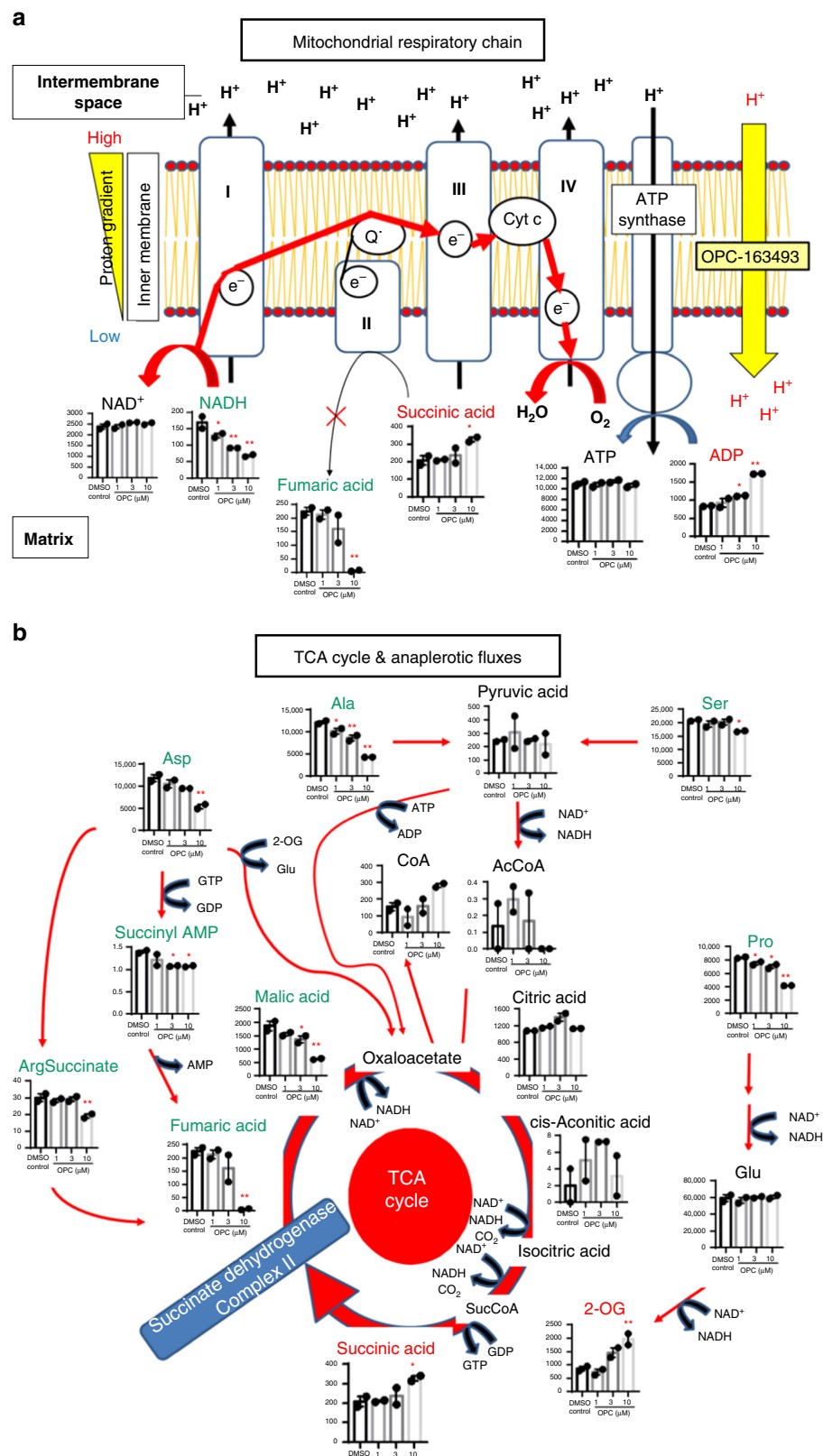

**Fig. 5** Metabolomic analysis of in vitro effects of OPC-163493 in HepG2 cells. Data represent mean ± SE ($n = 2$, *$P < 0.05$, **$P < 0.01$, OPC vs. DMSO control using a two-tailed Williams' test). **a** Effects on MRC. **b** Effects on the TCA cycle and anaplerotic fluxes. AcCoA acetyl CoA, ArgSuccinate argininosuccinic acid, Succinyl AMP adenylosuccinic acid

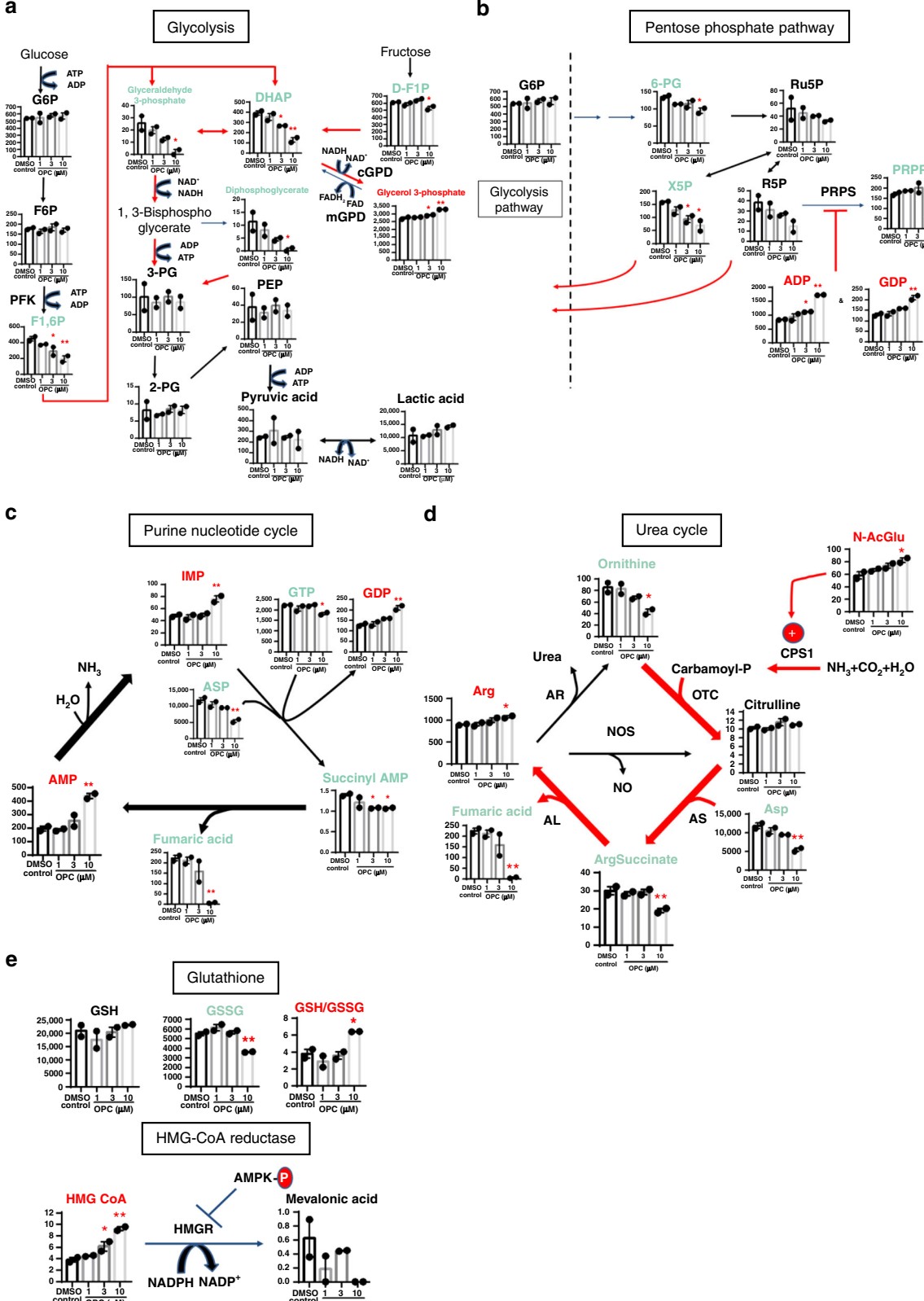

**Fig. 6** Metabolomic analysis of in vitro effects of OPC-163493 in HepG2 cells. Data represent mean ± SE ($n = 2$, *$P < 0.05$, **$P < 0.01$, OPC vs. DMSO control using a two-tailed Williams' test). **a** Effects on glycolysis. G6P glucose 6-phosphate, F6P fructose 6-phosphate, 3-PG 3-phosphoglyceric acid, 2-PG 2-phosphoglyceric acid, PEP phosphoenolpyruvic acid. **b** Effects on pentose phosphate pathway. 6-PG 6-phosphogluconic acid, Ru5P ribulose 5-phosphate, X5P xylulose 5-phosphate. **c** Effects on purine nucleotide cycle. **d** Effects on urea cycle. OTC mitochondria ornithine transcarbamoylase, AR arginase, NOS nitric oxide synthases. **e** Effects on glutathione redox and HMG-CoA reductase

HbA1c values with those of the "six-week vehicle-treated" group, aggressive progression of diabetes was observed during the treatment period (11–17 weeks of age). Of 116 target metabolites, 101 were detected by CE-QqQ/TOFMS analysis[32–34] (Supplementary Table 20).

First, principal component analysis (PCA) made it clear that three group animals were clearly segregated into the first principal

component (PC1) and the separation between baseline control and vehicle groups was remarkable, indicating the progression of DM over 6 weeks. OPC-163493 intervention reversed this finding in the direction of the baseline (Fig. 7a).

As ZDF rats are leptin receptor-deficient, hyperglycemic, and hyperphagic, their livers would chronically receive excessive nutritional input from the blood. Therefore, ATP, GTP, and

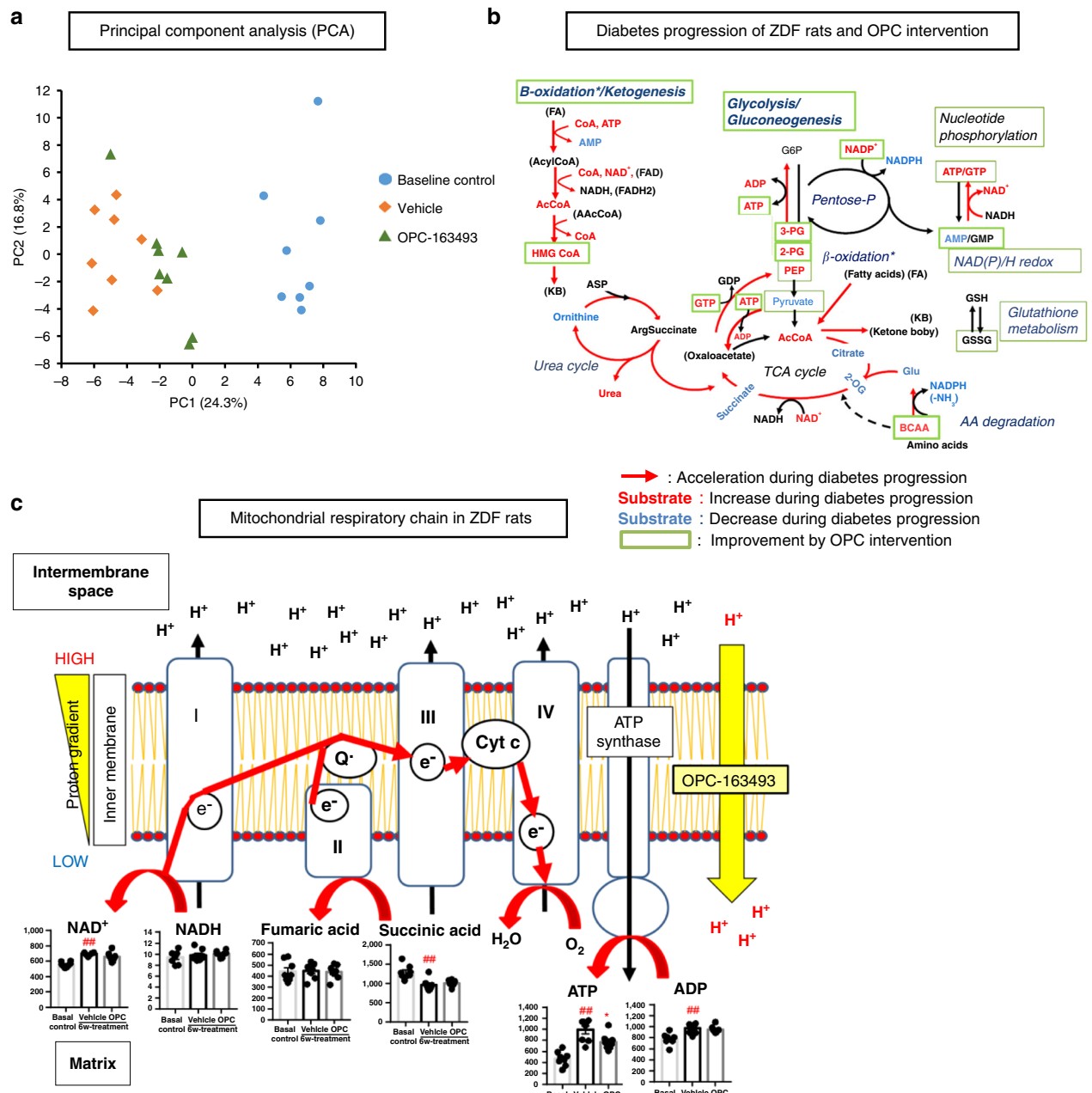

**Fig. 7** Metabolomic analysis of long-term and in vivo effects of OPC-163493 in ZDF rats. At baseline, liver samples were taken from the baseline control group of 11-week-old ZDF rats and the other animals were started on oral administration (BID) of OPC-163493 (6 mg kg$^{-1}$ day$^{-1}$) or its vehicle solution. After 6-week dosing (at the age of 17 weeks), liver samples were taken from both animals, and the liver metabolites of all three groups were then analyzed and compared. **a** PCA of metabolomics data comparing liver metabolites between baseline control (blue circle), vehicle 6-week-treated (orange diamond), and OPC-163493 6-week-treated (green triangle) groups of ZDF rats ($n = 8$). **b** Schema of diabetes progression and effects of OPC-intervention in the ZDF rats. Red arrows, substrates of red letters and blue letters represent accelerated pathways (imputed from the metabolite changes), significantly increased substrates, and decreased substrates, respectively, in the course of diabetes progression aged between 11 and 17 weeks in ZDF rats. Ameliorated pathways and substrates (significantly reversed to baseline) by OPC-intervention are enclosed in green boxes. AAcCoA Acetoacetyl CoA. **c** OXPHOS activation in the liver MRC along with diabetes progression of ZDF rats and uncoupling effect of OPC-163493. Data represent mean ± SE ($n = 8$, *$P < 0.05$, **$P < 0.01$, vehicle group vs. OPC treatment group using an unpaired $t$ test, #$P < 0.05$, ##$P < 0.01$, Baseline control group vs. vehicle group using an unpaired $t$ test). All units of vertical lines in metabolite graphs are nmol g-tissue$^{-1}$ (except ratios of substrates)

AcCoA were significantly elevated in vehicle-treated rats (Fig. 7b, Supplementary Fig. 5c, d). Accelerated OXPHOS in these vehicle-treated rat livers was indicated by a significant increment in the adenylate energy charge and NAD$^+$, and a decrement in AMP and succinate (Fig. 7c, Supplementary Fig. 5c, e). As expected, ATP was decreased and AMP increased in the OPC-163493-treated animals (Supplementary Fig. 5c). OPC-163493 treatment significantly raised the AMP/ATP ratio, according to the results of the in vitro study (Fig. 4a). On the other hand, stalling of SDH/Complex II activity, as observed in HepG2 cells (Fig. 5a), was not observed in this in vivo study (Fig. 7c). Other common features were observed in the glycolysis/gluconeogenesis pathway and glutathione metabolism (Fig. 7b, Supplementary Fig. 5f). OPC-163493 treatment induced acceleration of glycolysis, suppression of gluconeogenesis and improved oxidative stress[44].

OPC-163493 treatment generally altered the guanylate pool, NAD(P)/H redox[45], ketogenesis[46], and branched-chain amino acid (BCAA) catabolism[47], and their alterations were likely to push disease progression back to baseline levels (Fig. 7b, Supplementary Fig. 5e–g).

**Cardiovascular beneficial effects of OPC-163493**. To assess the effects of OPC-163493 on cardiovascular (CV) and cerebrovascular function, we performed an intervention study with OPC-163493 in salt-loaded stroke-prone spontaneously hypertensive rats (SHRSPs)[48]. In this study, stroke-related symptoms and deaths were observed. In the control group, median days

from salt loading to the first symptoms and death are 47.5 (95% CI, 43.0–48.0) and 54.0 (95% CI, 50.0–58.0), respectively. In contrast, OPC-163493 intervention significantly delayed the occurrence of both events: the median days were 55.5 (95% CI, 52.0–56.0) for the stroke-related symptoms and 63.5 (95% CI, 60.0–64.0) for death (Fig. 8a and Supplementary Table 21). We also measured blood pressure before the development of stroke. Six-week-old SHRSPs already showed moderate hypertension and salt loading induced further hypertension, attaining a systolic blood pressure of over 260 mmHg on day 41. OPC-163493 treatment slightly but significantly lowered systolic blood pressure on days 27 and 41 (Fig. 8b and Supplementary Table 21). To evaluate renal function, urinary collection was performed in metabolic cages. The poor condition of animals in the control group was reflected in significant appetite loss (Fig. 8c and Supplementary Table 21). While there was no difference in water (including 1% sodium chloride) consumption, urinary volume tended to reduce in the OPC-163493 treatment group (Supplementary Table 21). Urinary albumin-to-creatinine ratio was prominently improved by OPC-163493 intervention, lowering urinary albumin excretion by 68% from the value in the control group (Fig. 8d and Supplementary Table 21).

Next, in order to confirm the reproducibility of a blood pressure-lowering effect and a protective effect on renal function as well as their effective doses, another 6-week dosing study was carried out (second experiment). In this study, blood pressures were measured every 2 weeks, and urine samples were collected in metabolic cages for the last 24 h before sacrificing animals for

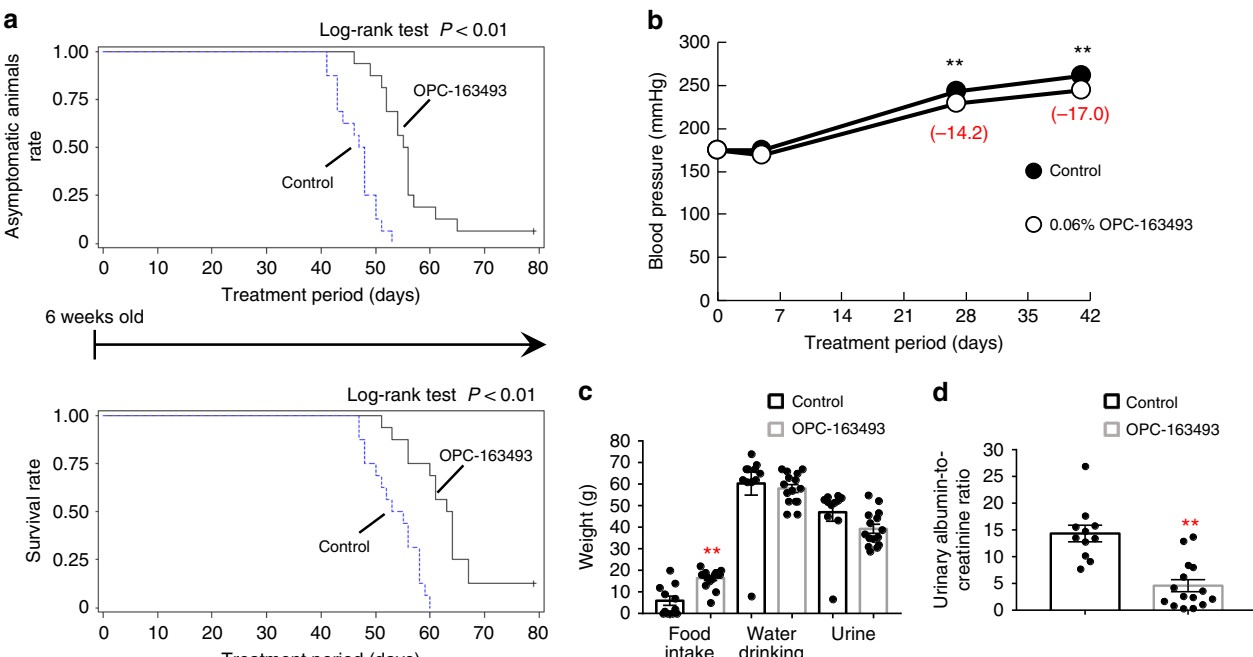

**Fig. 8** Effects of OPC-163493 in salt-loaded male SHRSPs. **a** Survival rates and rates for asymptomatic animals. Initial animal numbers of both groups are 16. Salt loading (1% sodium chloride in drinking water) and treatment (0.06% OPC-163493 mixed chow or control chow) commenced on the same day (day 0) in the animals at the age of 6 weeks. Significant delays were observed for both stroke onset (debility, convulsion, tremor, paralysis, rigors, cerebral edema, and/or bleeding) and death (P < 0.01, control vs. OPC treatment by log-rank test). **b** Blood pressure-lowering effect of OPC-163493 on salt-loaded SHRSPs. Systolic blood pressures (SBPs) are shown. Data represent mean ± SE (n = 16). As significant decreases were found in the overall treatment differences in SBP by the MMRM method (OPC vs. control chow group, P < 0.01), treatment-by-time interaction was then estimated by contrast-averaging the corresponding treatment differences at each time point using the MMRM method (**P < 0.01). Black circle, control; open circle, OPC-163493. **c** Effects of OPC-163493 on food intake, water drinking, and urinary volume of SHRSPs in metabolic cages between days 48 and 49 for 24 h. The number of animals surviving and tolerant to metabolic cages were 11 of 16 rats in the control group and 15 of 16 rats in the OPC treatment group. A significant reduction in food intake was observed in the control group. Data represent mean ± SE (control: n = 11, OPC-163493: n = 15, **P < 0.01 by unpaired t test). Black bar, control; silver bar, OPC-163493. **d** Effects of OPC-163493 on urinary albumin in SHRSPs. Data represent mean ± SE (control: n = 11, OPC-163493: n = 15, **P < 0.01 by unpaired t test). Black bar, control; silver bar, OPC-163493

blood collection. High doses (0.06%) of OPC-163493 significantly lowered blood pressure on day 28, but low doses (0.02%) failed to show a significant effect (Supplementary Fig. 6a and Supplementary Table 22). Animal deaths had already occurred by day 41, three (days 33, 35, 38) in the control group and two (days 35, 41) in 0.02% mixed chow group, but all animals in the 0.06% dose group survived. High-dose treatment again lowered urinary albumin excretion by 63% compared with the values in the control group. In addition, plasma creatinine, creatinine clearance, and BUN were all improved by the 0.06% dose of OPC-163493 (Supplementary Fig. 6b–e and Supplementary Table 22).

Taken together, OPC-163493 is expected to have a potency that lowers CV risk and prevents kidney deterioration. Plasma OPC-163493 concentration at 6 AM ($C_{6AM}$) of 0.06% chow in the first experiment on day 51 was $2.48\ \mu g\ mL^{-1}$ (7.4 $\mu M$) (Supplementary Table 21). $C_{6AM}$ of 0.02 and 0.06% chow in the second experiment on day 35 were 0.67 and $3.14\ \mu g\ mL^{-1}$ (2.0 and 9.4 $\mu M$), respectively (Supplementary Table 22). Those exposure levels were within the pharmacologically effective range of plasma concentrations in diabetic animal models in which antidiabetic effects of OPC-163493 were observed.

**The effect of OPC-163493 on the rat thoracic aorta**. To identify a possible mechanism of OPC-163493 on vascular function, we characterized its effect on constriction of the rat thoracic aortas induced by phenylephrine (PE), since Zhang et al. reported that carbonyl cyanide *m*-chlorophenylhydrazone (CCCP) directly induced vasorelaxation of SD rat aortas and the vasodilation was endothelium-independent[49]. We measured isometric tension changes induced by PE, both in intact and endothelium-rubbed vessels (Supplementary Fig. 7a). Treatment with CCCP relaxed PE-induced contraction of rat aortas dose-dependently, but its relaxation was irreversible; after high-dose treatment with CCCP, the vessels no longer responded normally to PE (Fig. 9a and

Supplementary Fig. 7b). In contrast, the influence on rat aortas up to 10 $\mu M$ of OPC-163493 was negligible (Fig. 9a and Supplementary Fig. 7c).

Next, to elucidate whether OPC-163493 affects nitric oxide (NO) bioavailability, we investigated the effects of OPC-163493 on relaxant activity induced by sodium nitroprusside (NP), an NO donor in rat thoracic aortas. When ascending doses of NP were added from 10 min after PE addition, NP treatment gradually relaxed arteries constricted by PE; the $EC_{50}$s (DMSO controls) were 36.1 (95% CI, 23.6–55.3) nM for intact aorta and 4.32 (95% CI, 3.24–5.76) nM for endothelium-rubbed aorta, respectively (Fig. 9b, Supplementary Fig. 7d and Supplementary Table 23). The far greater requirement for NP to relax the intact aorta was thought to be due to the endothelial barrier and NO attrition in endothelia. Pretreatment with OPC-163493 (5 $\mu M$) significantly enhanced the sensitivity of aortas against NP relaxation by 6.6-fold for intact aorta and 3.2-fold for endothelium-rubbed aorta, respectively, compared with DMSO treatment (Fig. 9b, Supplementary Fig. 7d and Supplementary Table 23). Since the enhancement was more prominent in the intact aorta with endothelia, this suggests that OPC-163493 may improve access of NO to vascular smooth muscle.

**Safety assessment of OPC-163493**. To demonstrate the safety of OPC-163494 sufficiently for clinical trials, we performed 4-week and 13-week repeated oral dose toxicity studies in male and female rats. The results are summarized in Supplementary Tables 24, 25, and 26. In the 4-week study, no observable adverse effect levels (NOAELs) of male and female rats were 30 mg kg$^{-1}$ (AUC24 h: 100 mg h mL$^{-1}$) and 100 mg kg$^{-1}$ (AUC24 h: 398 mg h mL$^{-1}$), respectively. However, toxic changes observed in males at the lowest observable adverse effect level (LOAEL: 100 mg kg$^{-1}$) were confined to abnormal hematological values and hypertrophy of hepatocytes, and these changes were slight. Decreased body weight gain

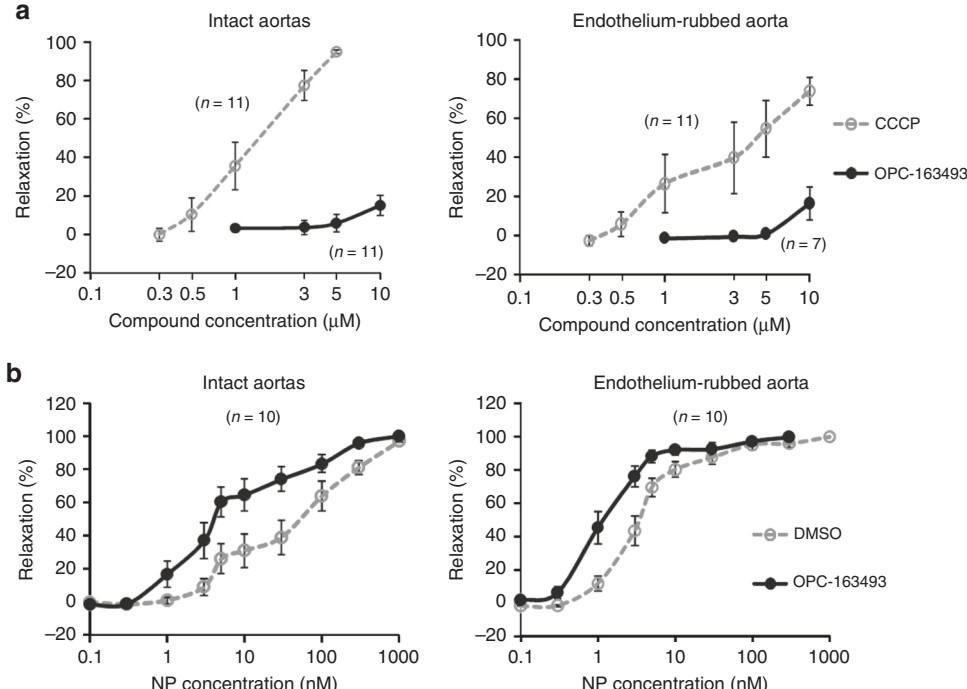

**Fig. 9** Effects of OPC-163493 on rat thoracic aortas. Data represent mean ± SE. **a** Effects of CCCP and OPC-163493 on PE-induced constriction of rat thoracic aortas. Test compounds were added 10 min after PE addition and their relaxation effects were measured for 10 min. Gray open circles and dashed line, CCCP; black circles and line, OPC-163493. **b** Sensitization of OPC-163493 to NP-induced relaxation of rat aortas. OPC-163493 was pretreated 10 min before PE addition and NP treatment was started 10 min after the PE addition. EC$_{50}$ values were estimated by regression analysis (NP concentrations were subjected to Log transformation). Gray open circles and dashed line, DMSO control; black circles and line, OPC-163493

was observed in both sexes of the 300 mg kg$^{-1}$ group and slightly increased body temperature was observed at 4 h after dosing in the males of the 300 mg kg$^{-1}$ group (Supplementary Tables 24 and 26). In the 13-week study, no toxicological change was observed at all setting doses (up to 100 mg kg$^{-1}$); in other words, extension of the experimental period did not worsen the toxicity of OPC-163493 (Supplementary Table 25).

## Discussion

OPC-163493 has a unique chemical structure which differs from that of any known mUncouplers. It exhibited insulin-independent antidiabetic and antisteatotic effects in multiple animal models (Fig. 10a). The MOA of OPC's blood glucose-lowering effect is considered to be an enhancement of cellular energy expenditure in the liver by inducing mUncoupling (Fig. 10b). On the other hand, OPC-163493 did not augment whole body energy expenditure, while lowering HbA1c. The reason is considered to be its unique PK profile: the main organs of its distribution after oral administration are liver and kidney alone. Since organ specificity was sharply restricted, we did not observe significant improvements in Rd (i.e. insulin-stimulated glucose disposal mainly in the muscle) in the clamped state of the hyperinsulinemic-euglycemic clamp test. OPC-163493, however, significantly lowered HGP at the basal state. Contrary to our expectation, OPC-163493-treatment showed only a tendency towards improvement in GIR and HGP under the clamp condition. The reason why significant differences were not detected in these parameters was thought to be due to the large data variation observed in GIR.

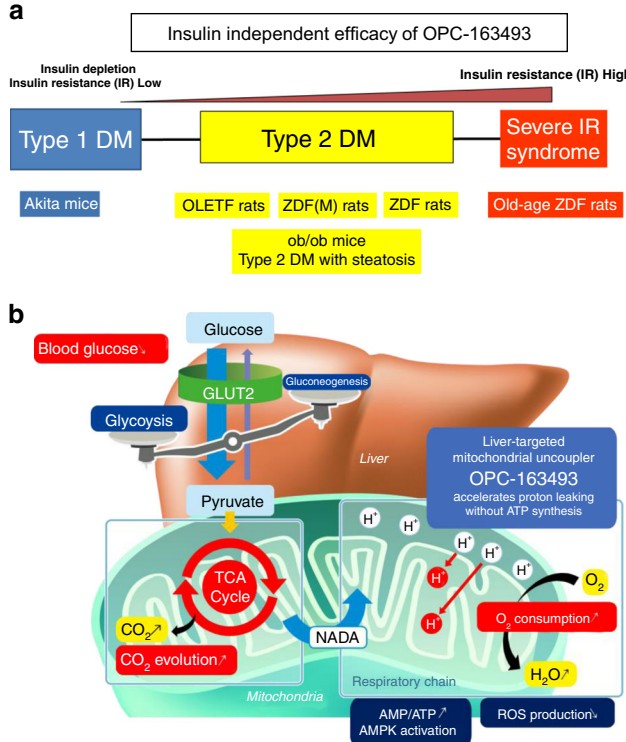

**Fig. 10** Pharmacological efficacy and MOA of OPC-163493.
**a** Pharmacologically effective animal models and expected target patient population of OPC-163493. OPC-163493 is expected to be applicable to a wide variety of hyperglycemic conditions besides type 2 DM, from type 1 DM to severe insulin resistance syndrome through its insulin-independent efficacy. **b** MOA of OPC163493 blood glucose-lowering effect. GLUT2 facilitated glucose transporter, type2/*SLC2A2*. This illustration was designed by N.K. and created by Heart-full Kawauchi Co. Ltd. (Tokushima, Japan) according to N.K.'s request as our original image

The localized exposure contributed to the liver-specific activation of OCR and its safety. Our preclinical toxicity study revealed that OPC-163493 had a remarkably wide safety range between NOAEL and the effective plasma concentrations (Supplementary Fig. 8).

Metabolomic analyses elucidated and implicated several aspects of mUncoupling. In vitro analysis showed interesting metabolic shifts to preserve Δψ counteracting the mUncoupling activity of OPC-163493: the electron flow from Complex I, via III, to IV was enhanced, but, in contrast, the input into SDH/Complex II (and also mGPD) was left in a highly uncoupled state. This could be explained by the fact that Complexes I, III, and IV but never Complex II assemble as supercomplexes, which have functional advantages (Supplementary Discussion 1)[50,51]. In an in vivo study, however, such drastic changes were not noted in the livers of hyperphagic ZDF rats abundant in ATP. We speculate that increased respiration through Complexes I, III and IV occurred as a cellular adaptation of HepG2 cells. OPC-163493 may encourage cells to use the most efficient catabolic (respiratory) mechanisms they can and promote optimal usage of available energy sources within the cells, to maintain Δψ. Depleting glucose from the HepG2 culture medium actually induced OPC-163493 to switch the metabolic flux from glycolysis to FAO drastically (Fig. 1i and Supplementary Fig. 1d), suggesting that treatment with OPC-163493 directly augments FAO.

OPC-163493 treatment caused a significant increment of the AMP/ATP ratio as expected, in both in vitro and in vivo studies. We also confirmed AMPK activation by OPC-163493 in HepG2 cells. Another common change was a significant increase in the GSH/GSSG ratio implying improvement of oxidative stress.

Although there have been some reports showing that chemical mUncouplers work as ROS generators, the ROS production was measured at an extremely high dose (e.g., FCCP: 5–100 μM)[52] in these studies. Maximum activities of FCCP in cells or isolated mitochondria were observed at submicromolar concentrations in our experience (Fig. 1e and Supplementary Fig. 1b). In the series of studies described above, we could demonstrate the ROS-reducing nature of OPC-163493: suppression of ROS production in isolated mitochondria, an increase in GSH/GSSG ratio in HepG2 cells and in the liver of ZDF rats, and reduction of oxidative stress markers in the plasma of rats treated long-term with OLETF.

We demonstrated that OPC-163493 treatment lowered blood pressure, delayed stroke-onset, extended survival, and ameliorated albuminuria in salt-loaded SHRSPs. These beneficial effects could be attributed to indirect AMPK activation induced by AMP elevation or direct ROS reduction in MRC by OPC-163493 treatment. The target tissue is considered to be vascular endothelia and/or the kidney because both tissues are profoundly associated with regulation of blood pressure and cardiovascular function. In addition, the kidney is another major organ to which OPC-163493 is distributed and the blood vessel wall is also constantly exposed to blood OPC-163493. Endothelial dysfunction is often characterized by increased ROS formation. The pathogenesis is often described as the vicious cycle of endothelial NO synthase uncoupling (Supplementary Discussion 2)[53]. OPC-163493 treatment may turn off the vicious cycle by improving NO bioavailability; we demonstrated that OPC-163493 indeed enhanced NO bioavailability in the rat thoracic aorta (Fig. 9b and Supplementary Fig. 7d). Another possible mechanism is AMPK activation in endothelia and/or the kidney. Endothelial activation of AMPK was reported to promote NO release and induction of prostacyclin (PGI2), to reduce ROS production, and to induce subsequent vasodilation[35]. Moreover, we investigated the possibility of off-target effects of OPC-163493 against 59 randomly chosen targets. There were no targets for which the percentage of

inhibition by OPC-163493 reached 50% at a concentration of 10 μM (Supplementary Table 27).

In summary, we demonstrated that OPC-163493 is expected to be effective in a wide variety of pathological conditions and that it may be useful not just in type 2 DM, but in conditions that range from type 1 DM to severe insulin resistance syndrome (Fig. 10a). In addition to the localized tissue distribution and good safety profile, expectation of CV benefits beyond glycemic control would encourage further investigation and development. The therapeutic concept of treating metabolic diseases with mUncoupling has long been taboo since the withdrawal of DNP. Future clinical trials of OPC-163493 are hoped to prove its efficacy and safety in humans, and to revive an old therapeutic concept.

## Methods

**Reagents, cells, and animals**. OPC-163493, 5-{5-methyl-2-[4-(trifluoromethyl) phenyl]thiazol-4-yl}-3H-[1,2,3]triazole-4-carbonitrile, white solid (AcOEt-Hexane); mp: 288.3−292.0 °C; $^1$H NMR (400 MHz, DMSO-$d_6$): δ 8.15 (d, $J = 8.1$ Hz, 2H), 7.90 (d, $J = 8.1$ Hz, 2H), 2.79 (s, 3H); $^{13}$C NMR (500 MHz, DMSO-$d_6$): δ 162.3, 144.6, 139.6, 136.0, 135.9, 130.1 (q, $J = 32.1$;Hz), 126.4, 126.3, 123.9 (q, $J = 272.3$ Hz), 118.4, 112.8, 12.4; IR (ATR): 2268, 1321, 1126, 1107, 1066 cm$^{-1}$; HRMS (ESI) $m/z$: [M−H]$^-$ calcd. for $C_{14}H_7F_3N_5S$, 334.0374; found, 334.0394; analysis (calcd., found for $C_{14}H_8F_3N_5S$): C (50.15, 49.95), H (2.40, 2.51), N (20.89, 20.83). Its deuterium-labeled OPC-163493 (all protons on the phenyl ring were deuterated) for the internal standard of a PK study were synthesized at Medicinal Chemistry Research Laboratories, Otsuka Pharmaceutical Co., Ltd., Tokushima, Japan. FCCP was also synthesized at Medicinal Chemistry Research Laboratories, Otsuka Pharmaceutical Co., Ltd. CCCP was purchased from Wako Chemical, Ltd. Phenylephrine hydrochloride, acetylcholine chloride, and sodium nitroprusside dihydrate were purchased from Tokyo Chemical Industry Co., Ltd., Wako Chemical, Ltd., and Sigma-Aldrich, respectively. [$^{14}$C]-OPC-163493 labeled at C(5) of 3H-[1,2,3]triazole-4-carbonitrile (specific radioactivity: 2.19 GBq mmol$^{-1}$) was synthesized at Curachem, Inc., Korea. DNP was purchased from Sigma-Aldrich.

CHO-K1 (#EC85051005) and HepG2 (#EC85011430) cells were purchased from DS Pharma Biomedical Co., Ltd. Control CHO and NaCT-CHO (Otsuka cell bank) were established by transfection of the pME18S (GenBank Accession no. AB009864) vector, pcDNA$^{TM}$3.1 (+) vector and their $NaCT$ gene inserted vectors into CHO-K1 cells, respectively and following G418 selection. When making cell stocks, cells were routinely tested for mycoplasma. After thawing of frozen cells, cell cultures were not maintained for a long duration (passages < 20).

Sprague−Dawley (SD) rats: Crl:CD(SD), ZDF rats: ZDF-Leprfa/CrlCrlj, ZDF (M) rats: ZDF(M)-Leprfa/CrlCrlj, and ob/ob mice: B6.V-Lep ob/J were purchased from Charles River Laboratories Japan, Inc. Akita mice: AKITA/Slc and SHRSPs: SHRSP/Izm were purchased from Japan SLC, Inc. OLETF rats: Otsuka Long-Evans Tokushima Fatty, and LETO rats: Long-Evans Tokushima Otsuka were purchased from Hoshino Laboratory Animals, Inc.

**Assay of the TCA cycle activation**. NaCT-CHO and control CHO were plated at 20,000 cells well$^{-1}$ into white CulturPlate$^{™}$-96 (PerkinElmer) 2 days before the assay. Prior to assay incubation, the culture plates were washed twice with washing buffer, 10 mM HEPES-Tris (pH 7.4) containing 140 mM choline chloride, 2 mM KCl, 1 mM CaCl$_2$, 1 mM MgCl$_2$. OPC-163493 was dissolved and diluted in DMSO (Wako Pure Chemical industries) to 20 mM, and further diluted to 40 μM in assay buffer, 10 mM HEPES-Tris (pH 7.4) containing 140 mM NaCl, 2 mM KCl, 1 mM CaCl$_2$, 1 mM MgCl$_2$. Each 25 μL of compound solution was serially diluted and added to wells of 96-well plates; subsequently 25-μL radio-labeled substrate solution containing 0.4 mM (0.4 MBq mL$^{-1}$) [1,5-$^{14}$C] citric acid (PerkinElmer) in assay buffer was added. After 1 h incubation at 37 °C, the reaction mixture was discarded and washed three times with prechilled washing buffer and then 0.1 mL MicroScint 20 (PerkinElmer) was added to each well. The radioactivity was measured using a TopCount (PerkinElmer). Nonspecific activity (NS cpm), total radioactivity (total cpm), and residual radioactivity (R cpm) were determined by counting of control CHO-plated wells, NaCT-CHO-plated wells without compound, and with various concentrations of compound, respectively. Percent of residual intracellular radioactivity was calculated by the equation:

$$\frac{[R - NS]}{[Total - NS]} \times 100 (\%). \quad (1)$$

Accordingly, diffusion of [$^{14}$C]-CO$_2$ was estimated by the equation:

$$\frac{[Total - R]}{[Total - NS]} \times 100 (\%). \quad (2)$$

The difference of [Total] and [$R$] disappeared in the presence of 0.1 μM antimycin A. In this assay, if not affected by antimycin A addition, the compound means for the NaCT inhibitor and its % of inhibition were calculated using Eq. (2).

**Mitochondrial isolation and mitochondrial swelling assay**. Mitochondria were isolated from the livers of male 8-week-old SD rats as follows[54]. After mincing a liver piece (approximately 1000 mg) by a razor on ice, it was collected with ice-cold liver isolation buffer (LIB) (250 mM sucrose, 10 mM Hepes, 0.5 mM EGTA, 1 mg/mL bovine serum albumin, pH 7.4) into a 15 mL conical tube. Following procedures were done at 4 °C. Liver pieces were centrifuged at $300 \times g$ for 1 min and the supernatant was discarded. The pellet was suspended with 12 mL of LIB and homogenized with a Downs homogenizer. The homogenate was centrifuged at $2000 \times g$ for 10 min and supernatant was collected. The supernatant was centrifuged again at $8000 \times g$ for 10 min and the pellet was resuspended with liver resuspension buffer (LRB) (250 mM sucrose, 10 mM Hepes, pH 7.4). This procedure was repeated again and the pellet was suspended with a small volume of LRB and the protein concentration of the mitochondrial suspension was measured using a Pierce BCA Protein Assay Kit (#23225, Thermo Fisher Scientific). Mitochondrial swelling assay was carried out as follows[17]. Isolated liver mitochondria were added to isotonic acetate buffer (5 mM Tris-HCl, 145 mM potassium acetate, 0.5 mM EDTA, 3 μM valinomycin, and 1 μM rotenone, pH 7.4) at a final concentration of 0.25 mg mL$^{-1}$ mitochondrial protein. Immediately after adding OPC-163493 and FCCP at the final concentrations indicated in Fig. 1b and Supplementary Fig. 1b, respectively, absorbance of the mitochondrial suspension at 600 nm was measured at a rate of 12 measurements min$^{-1}$ using a SpectraMax M2 Microplate Reader (Molecular Devices). Data were recorded with SoftMax Pro (version 5.4) software.

**Measurements of Δψ and mitochondrial ROS production**. In order to assess Δψ, a lipophilic dye Safranine O was used, in which the fluorescent intensity decreases upon Δψ-dependent distribution to mitochondria[55]. Isolated rat liver mitochondria were added at a final concentration of 0.25 mg mL$^{-1}$ mitochondrial protein to respiration buffer (5 mM HEPES, 135 mM Sucrose, 65 mM KCl, 5 mM KH$_2$PO$_4$, 2.5 mM MgCl$_2$, pH 7.4) supplemented with 10 μM Safranin O and 5 mM Succinate, as a substrate for the respiratory chain. After adding OPC-163493 at the concentrations indicated in Fig. 1c, the mitochondrial suspension was incubated for 15 min at room temperature and fluorescent intensity of Safranin O was measured using a SpectraMax M2 Microplate Reader with the 495/586 nm excitation/ emission wavelength pair. Relative changes in Δψ were calculated by changes in fluorescent intensity, assuming that the mitochondrial suspension treated with DMSO control had 100% Δψ and the mitochondrial suspension without Succinate had 0% Δψ. Mitochondrial ROS production was monitored by measuring H$_2$O$_2$ generated by mitochondria with Amplex UltraRed reagent[56] (#A36006, Thermo Fisher Scientific). Rat liver mitochondria were added to the respiration buffer described above, supplemented with 5 μM Amplex UltraRed and 1 U mL$^{-1}$ HRP, at a final concentration of 0.25 mg mL$^{-1}$ mitochondrial protein. Fluorescent intensity was measured with the 530/580 nm excitation/emission wavelength pair before and 15 min after adding OPC-163493; the amount of H$_2$O$_2$ produced during the treatment with OPC-163493 was calculated.

**Quantification of mUncoupling activity in HepG2 cells**. The mUncoupling activity of OPC-163493 and reference compounds was evaluated by OCR in the presence of oligomycin using an extracellular flux analyzer (Model Number: XF24-3, Seahorse Bioscience, Inc.). HepG2 cells were cultured in XF microplates coated with type I collagen at $6.5 \times 10^4$ cells well$^{-1}$ in 100 μL of the medium: MEM with Earle's Salts, L-Glutamine and non-essential amino acids, liquid, 10% fetal bovine serum (FBS), 1 mM sodium pyruvate solution, for 4–6 h. The cells were further cultured for 18 h or more after an additional 150 μL of medium was added. The cell culture medium was replaced with the assay solution: Krebs-Henseleit Mineral Solution (KHMS: 111 mM sodium chloride, 4.7 mM potassium chloride, 2 mM magnesium sulfate, 2 mM disodium hydrogen phosphate) with 25 mM D-glucose, 1 mM sodium pyruvate (pH 7.4); and incubated in an air incubator at 37 °C for approximately 30 min to reach equilibrium. The XF microplate was then set on XF, and after equilibration, a round of measurement which consisted of 6.5 min mixing, 0.5 min waiting, and a 2-min measuring period, was repeated three times to observe stable baseline values, followed by oligomycin, test or reference compound, and antimycin A injections in the order shown in Supplementary Fig. 9a. After each injection, a three-round measurement was conducted to examine the response of each compound. Each experiment measuring triplicate wells for test compound and control, and single well for reference compounds, was repeated five times.

**Measurements of cellular fluxes**. Cellular fluxes were evaluated using three metabolic indicators, ECAR, OCR, and CDER. ECAR and OCR were simultaneously measured using sensor cartridges for ECAR/OCR. Although sensor cartridges for CDER/OCR can measure both CDER and OCR, only CDER measurement was performed. The HepG2 cell culture, the composition of a round of measurement, and the medium replacement procedure for measurement of ECAR/OCR were the same as the procedure used for quantification of mUncoupling activity. To measure CDER, XF microplates were incubated in an air incubator at 37 °C for approximately 1 h; the culture medium was then replaced by the assay solution for OCR with 20 mM HEPES, and further incubated for about 30 min. The XF microplate was then set to XF, and after equilibration, a round of measurement was repeated three times in order to observe stable baseline

values, followed by injection of OPC-163493 or control. After injection, a three-round measurement sequence was conducted to examine the response evoked by the test compound (Supplementary Fig. 9b). Each experiment that measured wells in triplicate was repeated five times. To examine the effect of OPC-163493 in rat primary hepatocytes, Seahorse XFe96 Analyzer (Agilent Technologies) was used. Primary hepatocytes isolated from male SD rats (#HEP134-S) were obtained from Biopredic International, Inc. and seeded into XFe96 microplates at a density of $2 \times 10^4$ cells well$^{-1}$, according to the manufacturer's instruction. Assay conditions including assay solution formulation and measurement cycles were the same as the procedures used in HepG2 cells.

**Measurement of endogenous FAO.** HepG2 cells were seeded into XFe96 microplates at a density of $3 \times 10^4$ cells well$^{-1}$ and cultured overnight. The next day, culture medium was replaced with substrate-limited medium (DMEM containing 0.5 mM glucose, 1 mM GlutaMAX, 0.5 mM carnitine, and 1% FBS) and cells were cultured for a further 24 h. Prior to the assay, cells were washed and incubated with FAO assay medium (KHMS supplemented with 2.5 mM glucose, 0.5 mM carnitine, and 5 mM HEPES) for 30 min at 37 °C. For the Etomoxir (Eto) treatment, the assay medium was added with Eto at a final concentration of 40 μM, and incubated for 15 min at 37 °C. Measurements and injections of oligomycin, test compound, and antimycin A were performed as for the procedures used in mUncoupling activity measurement described above, using a Seahorse XFe96 Analyzer. To calculate FAO induced by each concentration of the test compound, the OCR elevation in the Eto-treated cells was subtracted from the OCR elevation seen in the cells not treated with Eto. FAO was measured in quadruplicate wells and repeated three times.

**Animal management.** Animals were housed under specific pathogen-free (SPF) conditions using a 12 h light and 12 h dark cycle with access to food and water ad libitum. MF (Oriental Yeast Co., Ltd.) was generally fed to animals except SD rats fed with CRF-1 (Oriental Yeast Co., Ltd.). In the case of HFD SD rats, high-fat chow (60 kcal% fat; Research Diet Inc., D12492) was fed starting 2 weeks before compound treatment. All MF chow containing OPC-163493 was prepared at Oriental Yeast Co., Ltd. All experiments were carried out in accordance with "Guidelines for Animal Care and Use at Otsuka Pharmaceutical Co., Ltd.". The study protocols were approved by the Animal Care and Use Committee at Otsuka Pharmaceutical Co., Ltd. The number of animals, which was likely to be the minimum required number, was carefully determined based on our preliminary experiments and previous experience after discussion with the Animal Committee. Each group allocation was performed using the stratified randomization method, on the basis of factors used for group allocation. The investigators were not blinded to allocation during experiments. Briefly, the protocols were as follows.

QWBA experiment: animal: male SD rat, 8 weeks old, $n = 5$, pregnant SD rats, 17 days of pregnancy, $n = 5$, factor for group allocation: body weight, dosing: 1 mg kg$^{-1}$ of [$^{14}$C]-OPC-163493 single oral administration (1 mg, 5 MBq/ 5 mL$^{-1}$ in 5% gum arabic solution), parameters determined: radioactivity in the tissues and blood.

OCR measurements in organ homogenate: animal: male SD rat, 11 weeks old, $n = 6$, factor for group allocation: body weight, dosing: 10 mg kg$^{-1}$ single oral administration (vehicle: 5% gum arabic solution), parameters determined: mitochondrial OCR in organ homogenates, plasma OPC-163493 concentration 2 h after dosing.

Six-week oral dosing study in ZDF rats: animal: male ZDF rat, 11 weeks old (baseline), $n = 16$, factors for group allocation: HbA1c, fasting blood glucose, body weight, dosing: 1, 2, 4, and 10 mg kg$^{-1}$ day$^{-1}$; 0.5, 1, 2, and 5 mg kg$^{-1}$ BID (vehicle: 5% gum arabic solution), dosing period: 6 weeks, parameters determined: HbA1c (change from the baseline), fasting blood glucose, fasting plasma insulin, body weight, food intake, PK parameters (Cmax, Tmax, AUC).

Six-week dosing study with mixed chow in Akita mice: animal: male Akita mice, 6 weeks old (baseline), $n = 12$, factors for group allocation: HbA1c, fasting blood glucose, body weight, dosing: 0.005, 0.01, and 0.02% mixed chow (comparator: control chow), dosing period: 6 weeks, parameters determined: HbA1c (value at the end of treatment), body weight, food intake, plasma OPC-163493 concentration change in a day (4 points 6 h intervals).

ITT in ZDF rats at the age of 21 and 25 weeks: animal: male ZDF rat, 21 or 25 weeks old, $n = 6$, factors for group allocation: fasting (3 h) blood glucose, body weight, dosing: 1 U kg$^{-1}$ insulin (Humulin R 100, Eli Lilly) i.p. (vehicle: saline), parameters determined: blood glucose (at 0, 15, 30, and 60 min).

Six-week oral dosing study in old ZDF rats: animal: male ZDF rat, 27 weeks old (baseline), $n = 6$, factors for group allocation: HbA1c, fasting blood glucose, body weight, dosing: 2, 6, and 10 mg kg$^{-1}$ day$^{-1}$; 1, 3, and 5 mg kg$^{-1}$ BID (vehicle: 5% gum arabic solution), dosing period: 6 weeks, parameters determined: HbA1c (change from the baseline), fasting plasma insulin, body weight, food intake, PK parameters (Cmax, Tmax, AUC).

Twenty-week dosing study with mixed chow in OLETF rats: animal: male OLETF rat, 27 weeks old (baseline), $n = 14$, normal control animal: male LETO rat, 27 weeks old (baseline), $n = 6$, factors for group allocation: HbA1c, fasting blood glucose, body weight, dosing: 0.01, 0.02, and 0.06% mixed chow (comparator: control chow), dosing period: 20 weeks, parameters determined: HbA1c (value at each point), body weight, food intake, oxidative stress markers (8-OHdG, 8-isoprostane), plasma OPC-163493 concentration at 6 AM on day 7.

Respiratory metabolism measurement study in ZDF(M) rats: animal: male ZDF (M) rat: The ZDF(M) rat was established as a type 2 DM animal model in the ZDF rat available from Charles River Laboratories Japan; its diabetic condition is milder than that of the ZDF rat, 7 weeks old (baseline), $n = 8$, factors for group allocation: HbA1c, body weight, dosing: 0.04% mixed chow (comparator: control chow), dosing period: 4 weeks, parameters determined: respiratory metabolism parameters (oxygen consumption [VO$_2$], carbon dioxide production [VCO$_2$], respiratory exchange ratio [RER], energy expenditure), locomotor activity, HbA1c (change from the baseline), body weight, food intake, plasma OPC-163493 concentration change in a day (4 points at 6 h intervals).

Ten-week dosing study with mixed chow in ob/ob mice: animal: male ob/ob mice, 8 weeks old (baseline), $n = 8$ except for control group ($n = 7$*), *Since one animal in the control chow group showed a 20% reduction in body weight with poor condition during the experiment, the animal was excluded and euthanized in accordance with the Otsuka's guidelines. Factors for group allocation: HbA1c, fasting blood glucose, body weight, dosing: 0.005, 0.01, and 0.02% mixed chow (comparator: control chow), dosing period: 10 weeks, parameters determined: HbA1c (change from the baseline), body weight, food intake, plasma and hepatic triglyceride, plasma OPC-163493 concentration at 6 AM following 10-week dietary administration.

Two-week oral dosing study in HFD SD rats: animal: male SD rat, 12 weeks old (baseline), $n = 18$, factors for group allocation: body weight, dosing: 2 and 4 mg kg$^{-1}$ day$^{-1}$; 1 and 2 mg kg$^{-1}$ BID (vehicle: 5% gum arabic solution), dosing period: 2 weeks, parameters determined: hepatic lipids profile, fasting blood glucose, fasting plasma insulin, plasma triglyceride, plasma total cholesterol, plasma adipokines (adiponectin, leptin, and resistin), body weight, food intake, and PK parameters (Cmax, Tmax, AUC).

Hyperinsulinemic-euglycemic study in ZDF(M) rats: animal: male ZDF(M) rat, 7 weeks old (baseline), $n = 12$, factors for group allocation: HbA1c, body weight, dosing: 0.04% mixed chow (comparator: control chow), dosing period: 3 weeks, parameters determined: basal and clamp hepatic glucose production rate (HGP), GIR, glucose disappearance rate (Rd), HbA1c (change from the baseline), blood glucose, plasma insulin, body weight, food intake.

Six-week oral dosing study in ZDF rats for metabolomic analysis: animal: male ZDF rat, 11 weeks old (baseline), $n = 8$, factors for group allocation: HbA1c, fasting blood glucose, body weight, dosing: 6 mg kg$^{-1}$ day$^{-1}$; 3 mg kg$^{-1}$ BID (vehicle: 5% gum arabic solution), dosing period: 6 weeks, parameters determined: HbA1c (value at the end of treatment), fasting blood glucose, PK parameters (Cmax, Tmax, AUC), concentrations of liver metabolic substrates.

Mixed chow dosing study in salt-loaded SHRSPs: animal: male stroke-prone spontaneously hypertensive rats, 6 weeks old (baseline), $n = 16$ (for first experiment) and $n = 10$ (for second experiment), factors for group allocation: blood pressure, body weight, dosing: 0.02% (for only second experiment) and 0.06% mixed chow (Comparator: control chow), salt loading: drinking water including 1% NaCl (from the age of 6 weeks), parameters determined: stroke-related symptoms (debility, convulsions, tremor, paralysis, rigors, cerebral edema, and/or bleeding), survival rate (for first experiment alone), systolic blood pressure, urinary albumin and creatinine, plasma creatinine (for second experiment alone), creatinine clearance (for second experiment alone), BUN (for second experiment alone), body weight, food and water intake in metabolic cages, plasma OPC-163493 concentration at 6 AM (on day 51 for first experiment or on day 35 for second experiment).

**PK experiments.** OPC-163493 concentrations in plasma of treatment group were determined by the high-performance liquid chromatographic-electrospray ionization tandem mass spectrometry (LC-ESI-MS/MS) method at the Department of Drug Metabolism and Pharmacokinetics, Nonclinical Research Center, Tokushima Research Institute, Otsuka Pharmaceutical Co., Ltd., Tokushima, Japan. To 50 μL of each plasma sample, 10 μL of methanol, 10 μL of the deuterium-labeled internal standard (IS) solution (2500 ng mL$^{-1}$), and 0.45 mL of methanol-formic acid (100:1, v/v) were added with mixing. The mixture was centrifuged to obtain the supernatant. The supernatant (0.2 mL) was mixed with 0.2 mL of purified water. The mixture was filtrated and the filtrate was used for the LC-MS/MS analysis. The plasma samples were diluted with blank animal plasma (blank plasma; anticoagulant: heparin sodium, supplier: Kitayama Labes Co. Ltd.) as necessary. Chromatographic separation of the analyte and IS was achieved using a SunShell PFP column (2.6 μm, 2.1 mm ID × 50 mm, ChromaNik Technologies Inc.). A binary gradient was formed with solvents A: ammonium formate aqueous solution (10 mM) and B: methanol at a flow rate of 0.35 mL min$^{-1}$. The ionization method was electrospray ionization (ESI) with a negative mode. Multiple reaction monitoring modes were employed utilizing the precursor and product ions of the analyte and IS.

**QWBA experiment.** Male SD rats ($n = 5$, 16 h fasted) and pregnant SD rats ($n = 5$, nonfasted) were administered 1 mg kg$^{-1}$ of [$^{14}$C]-OPC-163493. After 2, 8, 12, 24, and 72 h, each male and female rat was euthanized by inhalation of carbon dioxide. After tail and limb amputation, the hair of the carcass was sheared with an electric hair clipper, followed by two coatings of 0.5% carboxymethyl cellulose gel (CMC) and embedding in 4% CMC. The frozen blocks were stored at −15 °C or below. The frozen blocks of the animals and standard samples (0.1, 0.3, 1, 3, 10, 30, and 100 kBq g$^{-1}$ of rat blank blood; Japan SLC Inc., anticoagulant: heparin sodium) for the calibration curve were sliced at −20 °C with a thickness of 30 μm and freeze-dried

at −15 °C or below using a cryomicrotome (CM3600XP, Leica Microsystems KK). The sections covered with a protective film were exposed to the imaging plate (SR2040, Fujifilm Corp) for 4 days in a cartridge under lead protection. Evaluation of the sections was performed using the Fluoro Image Analyzer FLA-5100 (Fujifilm Corp) with scanner conditions (gradation, 65536; resolution, 50 μm). The radioactivity in the tissues was evaluated based on gray-scale intensity of whole body radioluminograms. The concentrations of radioactivity (ng-eq g$^{-1}$) in blood and tissues were determined using a calibration curve ($\log Y = a \log X + b$, $Y$: density measurement unit ([PSL-BG] mm$^{-2}$), $X$: concentration of radioactivity (ng-eq g$^{-1}$), PSL: photo-stimulated luminescence, BG: background) between radioactivity and regions of interest (ROIs) in the blood and tissues using the QWBA method. The determination coefficients of the calibration curves were from 0.9902 to 0.9962 (acceptable range: 0.98 and over). The tissue to blood ratio (T/B ratio) was calculated. The below-lower-limit of quantification (BLQ) was 20.00 ng-eq g$^{-1}$. The radioactivity of the radioluminogram was determined using Multi Gauge version 3.1 (Fujifilm Corp).

**Measurements of plasma and hepatic OPC-163493 concentrations.** Male SD rats (16 h fasted) were administered 1 mg kg$^{-1}$ of [$^{14}$C]-OPC-163493. After 1, 2, 4, 8, and 24 h ($n = 3$, each time point), a blood sample was collected from the abdominal vena cava under anesthesia with isoflurane. Following blood sampling, the animals were euthanized by exsanguination, and the liver was then collected. The blood was centrifuged at 1940 × $g$ for 15 min at 4 °C and a 50 μL sample was mixed with 5 mL of Insta-Gel Plus (PerkinElmer, Inc.). The liver was weighed, and homogenized with three times the volume of saline. An aliquot of the sample was weighed and added to 1 mL of Biomerit tissue solubilizer solution (Nacalai Tesque, Inc.). The mixture was treated with a sonicator (Sonicator Biomerit, Daiichi Pure Chemicals Co, Ltd.) for about 0.5 h, after which 15 mL of Hionic-Fluor (PerkinElmer, Inc) was added. The radioactivity was measured for 1 min using a liquid scintillation counter (LSC: LSC-6101, Aloka). The counting efficiency was corrected using the external standard source method. The concentration of radioactivity in plasma and liver was expressed as equivalents of OPC-163493 (μg eq mL$^{-1}$ or μg eq g$^{-1}$). The pharmacokinetic parameters (Cmax, Tmax, and AUC24 h) of the radioactivity were calculated from the mean radioactivity concentration in each of the plasma and liver samples by noncompartmental model analysis using Phoenix WinNonlin Version 6.3 (Pharsight Corp).

**OCR measurement of organ homogenate samples from SD rats.** According to the instruction manual, OCR measurements were performed using an Oxytherm system (OXYT-1) and its software, Oxygraph (Hansatech Instruments). Two hours after oral administration, plasma samples (heparinized) were taken from the inferior vena cava of OPC-treated animals under isoflurane anesthesia for measurement of its plasma concentration. Immediately after euthanasia by exsanguination, a piece of liver from the left lateral lobe, pieces from the triceps surae muscle (the soleus, medial, and lateral gastrocnemius, 1:6:6 mixture[57]) from the right leg, and the right brain hemisphere (except the olfactory bulb) were extracted from each animal. Organ samples (the skeletal muscle was pre-minced) were homogenized (1000 rpm × 10 strokes, once for the liver and brain, 5000 rpm × 5 strokes, three times for the skeletal muscle) with three times the volume of homogenate buffer: 10 mM HEPES with 250 mM sucrose, 0.5 mM EGTA and 1 mg mL$^{-1}$ BSA (fatty acid free) for the liver and skeletal muscle, 5 mM HEPES with 225 mM mannitol, 75 mM sucrose, 0.5 mM EGTA and 1 mg mL$^{-1}$ BSA (fatty acid free) for the brain, equivalent to the wet weight, using a Potter homogenizer on ice and centrifuged (300 × $g$, 4 °C, 1 min for the liver and brain, 2000 × $g$, 4 °C, 1 min for the skeletal muscle). The electrode chamber was filled with 850 μL respiration buffer: 5 mM HEPES with 135 mM sucrose, 65 mM potassium chloride, 2.5 mM magnesium chloride, 5 mM potassium dihydrogenphosphate (pH 7.4) and the plunger was inserted. After stabilization of the voltage, measurements began and were repeated every second. A 150 μL volume of homogenized sample was added to the chamber, and 2 min after this addition, 1 mM Antimycin A solution (10 μL for the liver, 5 μL for the skeletal muscle and brain) was added. Sixty seconds of measurement from 60 to 120 s after addition of the sample, and 30 s of measurement from 60 to 90 s after the addition of Antimycin A solution represented whole OCR and nonmitochondrial OCR, respectively. (Oxygraph software automatically drew a line of best fit between the two time points specified, and the line of best fit was calculated by least squares regression). Therefore, mitochondrial OCR was calculated by subtracting nonmitochondrial OCR from total OCR. Each measurement of three organs was duplicated and the mean was calculated. Protein concentrations of homogenized samples were measured using Protein Assay Dye (Bio-Rad), and mitochondrial OCRs were corrected for the protein concentration.

**Measurement of blood glucose, plasma insulin, and HbA1c.** Blood for blood glucose measurement was taken from the tail vein without anesthesia. Plasma samples of heparinized blood taken from the inferior vena cava under isoflurane anesthesia were used for insulin measurement. Blood glucose levels and insulin levels in plasma were determined by Glutest Mint (Sanwa Kagaku Kenkyuusho) and Ultra Sensitive Rat Insulin ELISA Kit (Morinaga Institute of Biological Science), respectively. Heparinized blood samples for baseline HbA1c and efficacy measurements were taken from the tail vein without anesthesia, and the inferior

vena cava under isoflurane anesthesia, respectively. HbA1c values were determined using a DCA Vantage Analyzer (Siemens).

**Measurement of oxidative stress markers.** Plasma samples were acquired from heparinized blood taken from the inferior vena cava under isoflurane anesthesia. Before 8-OHdG measurement, samples were filtrated using an ultrafiltration membrane (AmiconUrtra 10KDa, Millipore). Plasma 8-OHdG levels were measured using a Highly Sensitive ELISA Kit for 8-OHdG (Japan Institute for the Control of Aging, NIKKEN SEIL Co., Ltd.). Before 8-isoprostane measurement, samples were hydrolyzed and treated with solid phase extraction (SPE) purification according to the instructions. Plasma 8-isoprostane levels were determined using an 8-isoprostane EIA Kit (Cayman Chemical Company).

**Measurement of respiratory metabolism.** Oxygen consumption [VO$_2$], carbon dioxide production [VCO$_2$], respiratory exchange ratio [RER], and energy expenditure were measured for about 26 h under the fed state with an Oxymax Equal Flow System (Columbus Instruments, Columbus, OH USA). Twenty-four-hour data were used for evaluation after 1.5 h of measurement during a warm-up period. At the same time, diurnal locomotor activity was measured with an ACTIMO system (Shinfactory Inc.). Data were acquired every 10 min. The measurement was performed by putting the animals in chambers (one animal per chamber). RER and energy expenditure was calculated from the following equation[58,59]:

$$RER = VCO_2/VO_2, \tag{3}$$

$$\text{Energy expenditure} = CV \times VO_2, CV = 3.815 + 1.232 \times RER. \tag{4}$$

This experiment was performed in the Kumamoto Laboratory, LSI Medience Corporation, Kumamoto, Japan.

**Measurement of plasma TG, cholesterol, and adipokines.** Plasma samples (heparinized) were acquired from the inferior vena cava under isoflurane anesthesia. Triglyceride and total cholesterol were measured using Daiyacolor·LiquidTG-S (TOYOBO CO., LTD) and L-Type Wako CHO·M (Wako Pure Chemical Industries, Ltd.), respectively. Adiponectin, leptin, and resistin were quantitated using the Mouse/Rat Adiponectin ELISA kit (Otsuka Pharmaceutical Co., ltd.), Rat Leptin ELISA Kit (#YK050, Yanaihara Institute Inc.), and ELISA Kit for Resistin (#SEA847Ra, Cloud-Clone Corp.), respectively.

**Measurement of hepatic triglyceride.** After euthanasia by exsanguination, liver pieces were taken from the left lateral lobe. Lipid extraction from the liver piece was performed as follows: After adding three times the volume of saline (v/wet weight) and homogenization, the 50 μL of homogenate was added to 1 mL of Folch reagent (chloroform/methanol = 2/1). After vortexing for 5 s or more, the whole mixture was agitated overnight in an orbital shaker at room temperature. The mixture was centrifuged (20,000 × $g$, room temperature) and the supernatant was then dried. The residue was vigorously suspended with 50 μL of Folch reagent and 10 μL of MeOH/Triton X-100 (1/1), and dried again. Before measurement, the residue was re-dissolved in 95 μL of 10 mM HEPES buffer and sonicated. Triglyceride was measured using Daiyacolor·LiquidTG-S (TOYOBO CO., LTD).

**Lipidomic analysis.** We performed a quantitative and targeted metabolomic approach based on liquid chromatography-high-resolution mass spectrometry (LC-HRMS) analysis using the AbsoluteIDQ p400 kit (BIOCRATES Life Sciences AG, Innsbruck, Austria), according to the manufacturers' instructions. Briefly, after euthanasia by exsanguination under isoflurane anesthesia, a piece of liver was taken from the left lateral lobe, weighed at approximately 50 mg, and immediately frozen in liquid nitrogen. The tissue was homogenized with ethanol/phosphate buffer using Shake Master Neo (Biomedical Science Co., Ltd., Tokyo, Japan) and the homogenates were then centrifuged at 5000 × $g$ at 4 °C for 5 min. The supernatant (10 μL) was transferred onto the filter in a 96-well kit plate, already containing the relevant internal standards. After drying the filter under a stream of nitrogen, analytical targets were extracted with 5 mM ammonium acetate in methanol. The extracts were diluted with LC-HRMS mobile phase, and 20 μL of the final solution was subjected to flow injection analysis (FIA) using an LC-HRMS system that was configured with an Ultimate 3000 UPLC (Thermo Fisher Scientific, San Jose, CA, USA) coupled to a QExactive plus (Thermo Fisher Scientific, San Jose, CA, USA) with electrospray ionization in positive mode. The MetIDQ software included in the p400 kit was used for data evaluation and quantification of metabolite concentrations.

**Metabolomic analyses.** Metabolome measurements were carried out through a facility service at Human Metabolome Technologies, Inc., Tsuruoka, Japan.

Sample preparation for in vitro study: HepG2 cells (1 × 10$^7$ cells) were seeded into a 100 mm dish the day before the assay and cultured in MEM with Earle's Salts, L-Glutamine and non-essential amino acids, liquid, 10% FBS, and 1 mM sodium pyruvate solution. OPC treatment (DMSO control, 1, 3, or 10 μM) was performed for 30 min in FBS-free DMEM with high glucose (25 mM). The culture medium was aspirated from the dish and cells were washed twice using 5% mannitol solution (10 mL first and then 2 mL). The cells were then treated with

800 µL of methanol and left to rest for 30 s to inactivate enzymes. Next, the cell extract was treated with 550 µL of Milli-Q water containing internal standards (H3304-1002, Human Metabolome Technologies, Inc., Tsuruoka, Japan) and left to rest for another 30 s. The extract was obtained and centrifuged at $2300 \times g$ and 4 °C for 5 min and then 800 µL of upper aqueous layer was centrifugally filtered through a Millipore 5-kDa cutoff filter at $9100 \times g$ and 4 °C for 120 min to remove proteins. The filtrate was centrifugally concentrated and resuspended in 50 µL of Milli-Q water for CE-MS analysis.

Sample preparation for in vivo study: After euthanasia by exsanguination under isoflurane anesthesia, a piece of liver was taken from the left lateral lobe, weighed at approximately 50 mg, and immediately frozen in liquid nitrogen. The frozen sample was plunged into 1500 µL of 50% acetonitrile/Milli-Q water containing internal standards (H3304-1002) at 0 °C in order to inactivate enzymes. The tissue was homogenized three times at 1500 rpm for 120 s using a tissue homogenizer (Micro Smash MS100R, Tomy Digital Biology Co., Ltd., Tokyo, Japan) and the homogenate was then centrifuged at $2300 \times g$ and 4 °C for 5 min. Subsequently, 800 µL of upper aqueous layer was centrifugally filtered through a Millipore 5-kDa cutoff filter at $9100 \times g$ and 4 °C for 120 min, to remove proteins. The filtrate was centrifugally concentrated and resuspended in 50 µL of Milli-Q water for CE-MS analysis.

**Western blotting**. HepG2 cells were treated with serum-free DNEM with low glucose (5.5 mM) containing OPC-163493, FCCP, and metformin at the concentration indicated in Fig. 4b for 90 min. After treatment, cells were lysed in buffer (20 mM Tris-HCl, pH 8.0, 1% Triton-X 100, 1 mM EDTA, 1 mM EGTA) supplemented with protease inhibitor cocktail and phosphatase inhibitor cocktail (#04080-24 and #07574-61, respectively, Nacalai Tesque, Inc.). After removing cell debris by centrifugation at $14,000 \times g$ for 15 min at 4 °C, the resulting supernatant was subjected to western blotting analysis. A Pierce BCA Protein Assay Kit (#23225, Thermo Fisher Scientific) was used to measure protein concentrations. Protein samples (15 µg) were separated on a 4–15% gradient SDS–PAGE gel and transferred to a PVDF membrane. After blocking membrane with ECL Prime Blocking Reagent (RPN418, GE Healthcare Life Sciences), the membrane was incubated overnight at 4 °C with primary antibodies, as described below. The following primary antibodies were purchased from Cell Signaling Technology and used at a dilution of 1:1000: AMPK-α (#2532), Phospho-AMPKα (Thr172) (#2531), Acetyl-CoA carboxylase (#3662), and Phospho-Acetyl-CoA carboxylase (Ser79) (#3661). After incubation with the secondary antibody (anti-rabbit IgG, HRP-linked, #7074 Cell Signaling Technology) at a dilution of 1:2000 for 1 h at room temperature, blots were detected with ECL Prime Western Blotting Detection Reagent (#RPN2232, GE Healthcare Life Sciences). An ImageQuant LAS 4000 System (GE Healthcare Life Sciences) was used for signal detection. Protein expression levels were quantified with ImageQuant TL Software. The full-length blots are available in Source Data file worksheet "Fig. 4b".

**Hyperinsulinemic-euglycemic clamp test**. The hyperinsulinemic-euglycemic clamp test basically followed the reported procedure[60–62]. [3-³H]-glucose solution (37 MBq ml⁻¹, PerkinElmer) was diluted to 4.93 MBq mL⁻¹ for bolus injection and 740 kBq mL⁻¹ for continuous infusion with saline. Insulin (Humulin R 100, Eli Lilly) and somatostatin-14 (Bachem Ag) were diluted 1 U kg⁻¹ mL⁻¹ and 300 µg kg⁻¹ mL⁻¹, respectively, with saline containing 0.25% BSA (bovine serum albumin). Surgical catheterization was performed under anesthesia (5% isoflurane). After shaving hair from the surgical sites—the posterior and anterior neck of the rat—the sites were disinfected with 70% ethanol. Small horizontal and vertical incisions were made at these sites. Through the anterior opening, the right jugular vein was exposed with a tiny cut, followed by placement of an indwelling catheter (PE-50, Fisher Scientific) into the vein. The other end of the catheter emerged from the posterior neck incision subcutaneously. After closing the incisions, the surgical sites were disinfected with dilute iodine tincture. On recovery from anesthesia each rat was individually accommodated in a cage and reared for a further 2 days. On the day of the clamp test, rats were weighed and food and water were removed for 4 h before starting an insulin infusion (at −240 min). They were put into a free-moving system (SUGIYAMA-GEN Co., Ltd., Tokyo, Japan) and the jugular vein catheter was connected to three syringes, each containing one of [³H]-glucose solution, 20% glucose solution, or insulin and somatostatin mixed solution. After measuring blood glucose levels at −60 min, 444 kBq of [³H]-glucose solution was administered by bolus injection, followed by continuous infusion at a rate of 3.7 kBq min⁻¹. At 0 min, the mixed infusion of insulin and somatostatin was begun and 10 min later the mixed solution was continuously infused at rates of 10 mU kg⁻¹ min⁻¹ and 3 µg kg⁻¹ min⁻¹, respectively. With measurement of blood glucose levels every 3 or 4 min except during the first 10 min, a 20% glucose infusion was administered to target and clamp the blood glucose level at 140 mg dL⁻¹ by adjusting the GIR. Plasma samples for the determination of plasma [³H]-glucose specific activity were collected every 10 min from −20 to 0 min and from 60 to 120 min. The rate of glucose disappearance (Rd) and HGP were calculated in accordance with following formulae.

$$\text{Rd} (\text{mg kg}^{-1} \text{min}^{-1}) = \frac{[^3\text{H}]\text{glucose infusion rate} (\text{dpm min}^{-1})}{\text{Plasma} [^3\text{H}]\text{glucose} (\text{dpm mg}^{-1}) \times \text{Body weight (kg)}} \quad (5)$$

$$\text{HGP} (\text{mg kg}^{-1} \text{min}^{-1}) = \text{Rd} (\text{mg kg}^{-1} \text{min}^{-1}) - \text{GIR} (\text{mg kg}^{-1} \text{min}^{-1}). \quad (6)$$

**Measurement of blood pressure**. Blood pressures were measured by the indirect tail cuff method using an MK-2000ST blood pressure monitor (Muromachi Kikai Co., Ltd., Tokyo, Japan), without heating and averaged from three consecutive readings[63]. Based on our previous data, identifying an effect size with a difference of 15 mmHg SBP requires an estimated group size of $n = 15$ per group (Power = 0.8, unpaired $t$ test).

**Measurements of albumin, creatinine, BUN**. Urine was collected in a metabolic cage (one animal per cage) for 24 h. Heparinized blood was taken from the inferior vena cava under isoflurane anesthesia. Albumin, creatinine (cr), and BUN were determined by Albumin, Rat Urine, ELISA Kit (Exocell, Inc., PA, USA), Nephrat II, L-Type Wako CRE·M and UN (Wako Pure Chemical Industries, Ltd., Japan), respectively. Creatinine clearance (Ccr) was calculated as follows:

$$\text{Ccr} (\text{mL min}^{-1} 100\text{g BW}^{-1}) = \frac{\text{urinary cr} (\text{mg dL}^{-1})}{\text{plasma cr} (\text{mg dL}^{-1})}$$
$$\times \frac{\text{urinary volume} (\text{mL 24h}^{-1})}{24 \times 60 (\text{min})} \times \frac{100}{\text{BW (g)}}. \quad (7)$$

**Isometric tension measurement in the rat thoracic aorta**. Experiments were carried out according to Zhang's report[49]. Rat thoracic aortas were prepared from SD rats (9–13 weeks old). Thoracic aortic rings were fixed in baths filled with 5 mL of physiological salt solution (PSS: 130 mM NaCl, 4.7 mM KCl, 1.17 mM MgSO₄, 1.17 mM KH₂PO₄, 5.5 mM D-glucose, 1.6 mM CaCl₂, 14.9 mM NaHCO₃, pH 7.4) that was continuously aerated with gas (95% O₂ + 5% CO₂). Isometric tensions were measured using a UFER Easy Magnus System (UC-5A, Iwashiya Kishimoto Medical Instruments) equipped with isometric pressure transducers (UL-20GR, Minebea Co. Ltd). The tension values were recorded at one sec intervals and the average was calculated every 10 sec for use in analysis and graph-plotting.

**Preclinical toxicity study in rats**. Nonclinical safety (GLP compliant) studies were performed at Department of Drug Safety Research, Nonclinical Research Center, Tokushima Research Institute, Otsuka Pharmaceutical Co., Ltd, Tokushima, Japan.

**Examination of off-target effects of OPC-163493**. These assays were outsourced and carried out at Drug Development Solutions Center, SEKISUI MEDICAL Co., Ltd., Ibaraki, Japan.

**Statistical analysis**. Sample size was estimated based on our preliminary experiments and previous data; these data supported the assumptions of normal distribution and homoscedasticity. All analyses were performed using SAS Software (release 9.3, SAS Institute Japan). The level of significance was two-tailed with $P < 0.05$. Every statistical test was determined after consultation with the statistician in charge of statistics at Otsuka. Methods used for analysis are described in the figure and supplementary figure legends.

**Reporting summary**. Further information on research design is available in the Nature Research Reporting Summary linked to this article.

## Data availability

All data relevant to the manuscript are available from the corresponding author upon reasonable request. The data that support the findings of this study are presented in this published article, its supplementary information, source data file and public databases*. The source data underlying Figs. 1c, e−j, 2a−i, 3a−f, 4b, 8b−d and 9a, b, Supplementary Figs. 1a, c, d, 2a, c, 3a−n, q−w, 4, 5a, b and 6a−e and Supplementary Tables 3–26 are provided as a Source Data file. *Metabolomics data were deposited in the Metabolomics Workbench or the MetaboLights. Accession codes: ST001135, ST001136, and MTBLS836. A reporting summary for this article is available as a supplementary information file.

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

## Acknowledgements

We thank T. Sudo and K. Maeda for recommending the submission of this report. We also thank K. Nagano and H. Kawasome for reviewing and approving our study reports. We are grateful to K. Kakumoto for statistical instruction and support. We are also grateful to members of the Nonclinical Research Center, Tokushima Research Institute, Otsuka Pharmaceutical Co., Ltd. for experimental support.

## Author contributions

N.K. prepared the concept, organized most of the experiments, analyzed the data and composed the manuscript. T.O. designed and conducted the majority of animal experiments, and amply contributed to the interpretation of the data. K.T. and H.M. designed and conducted all the in vitro studies associated with mUncoupling activity. T.S. conducted all the PK studies. Y.H. carried out the lipidomic analysis. M.A. contributed to all animal experiments. T.B. performed the assay of the TCA cycle activation and supported all animal experiments. Y.K. and H.A. performed OCR measurements in organ homogenates. Y.I. conducted the preclinical toxicity studies. The first synthesis of OPC-163493 was done by M.I. S.S. was responsible for organic synthesis of cyanotriazole derivatives and the design direction.

## Additional information

**Competing interests:** All authors are employees of Otsuka Pharmaceutical Co., Ltd..

