## [Peer Review File · Nature Communications]

Reviewers' comments:

Reviewer #1 (Remarks to the Author):

In this study Kanemoto et al. identified OPC-163493 as a novel liver-localized mitochondrial uncoupler and evaluated its potential use for the treatment of T2D. Specifically, they demonstrate that oral administration of OPC-163493 significantly lowered Hba1c in multiple animal models of diabetes including ZDF rats (T2D), Akita mice (T1D), OLETF rats (late-onset T2D), and ob/ob mice (severely obese T2D) independent of changes in body weight. To further characterize the mechanism for OPC-163493's anti-diabetic effect, the authors went on to perform metabolomic analysis in HepG2 cells and ZDF rats after OPC treatment. Consistent with its role as a mitochondrial uncoupler, OPC treatment increased the AMP/ATP ratio and activated AMPK. OPC treatment also increased the GSH/GSSG ratio, indicative of reduced oxidative stress. Lastly, Kanemoto et al. assessed the effects of OPC-163493 on CV function and found that OPC treatment lowered blood pressure, extended survival and ameliorated albuminuria in salt-loaded spontaneous hypertensive rats.

Comments

Overall, this is an interesting and timely manuscript that describes the identification of a liver-directed mitochondrial uncoupler, OPC-163493, and its potential use as a novel therapy for diabetes and CVD. However, the experiments do not go far enough in demonstrating the mechanism by which this mitochondrial uncoupler ameliorates diabetes and improves CV function in the multiple animal models that were tested. The manuscript would benefit greatly from a more careful characterization of the anti-diabetic effects of OPC in vivo. Specific issues are outlined below:

- 1) The authors demonstrate that OPC-163493 is a direct mitochondrial uncoupler by measuring OCR in HepG2 cells in the presence of oligomycin (Figure 1d-e). Given the subsequent in vivo studies, does OPC increase OCR in primary rat or mouse hepatocytes?
- 2) While OPC accumulates primarily in the liver (Figure S2b), a significant amount of OPC also gets into the kidney. Markers of renal toxicity (BUN and creatinine) should be assessed in the diabetic models treated with OPC.

3) Mitochondrial uncouplers are expected to increase lipid oxidation and reduce intracellular lipid content. However, the authors only demonstrate that hepatic triglycerides are reduced in ob/ob mice after OPC treatment (Figure 2k). Does OPC treatment reduce liver triglycerides in the other animal models of T2D, such as ZDF rats?

4) The authors measure plasma concentrations of OPC and use QWBA to measure relative levels of OPC in vivo (Figure S2). It would be instructive to quantitate actual levels of OPC to determine what concentration of hepatic OPC is needed to produce anti-diabetic effects in vivo.

5) The effects of OPC on HbA1c and plasma glucose levels in ZDF rats and ob/ob mice (Figure S3a,s) are relatively modest compared to the effects observed in previous studies of mitochondrial uncouplers (Perry et al. Cell Metabolism 2013, Tao et al. Nature Medicine 2014, Perry et al. Science 2015, Abulizi et al. FASEB J. 2017). Moreover, OPC failed to significantly reduce plasma insulin concentrations in the models that were tested. These studies should be discussed in the context of the present study

6) Metabolic cage studies were only completed over the course of 26 h- this should be extended to 2-3 days to give animals time to acclimate to the metabolic cage system, as well as track the metabolic impact of OPC treatment during the light/dark cycles.

7) The authors suggest that the anti-diabetic effects of OPC are due to alterations in the hepatic metabolome. Given that mitochondrial uncouplers have previously been found to improve whole-body glucose metabolism by improving liver and skeletal muscle insulin sensitivity (Perry et al. Science 2015 and Tao et al. Nature Medicine 2014), it would strengthen the manuscript if the authors performed hyperinsulinemic-euglycemic clamp studies to directly assess tissue-specific insulin responsiveness in these models in vivo.

8) In Figure 2k, the authors demonstrate that OPC treatment significantly reduces hepatic triglycerides. Given the potential role of hepatic ceramides, diacylglycerols, acylcarnitines in mediating hepatic insulin resistance, the authors should measure these metabolites in liver and skeletal muscle. The authors should also assess the effects of OPC treatment on plasma adipokines/cytokines (adiponectin, leptin, resistin, IL-6, TNF-a).

9) No differences in plasma triglycerides were observed in ob/ob mice treated with OPC (Figure 2j). Did OPC treatment alter plasma triglycerides in any of the other diabetic

models? Was total plasma cholesterol or LDL-cholesterol altered?

10) The authors demonstrate that OPC treatment significantly increases pAMPK and pACC in HepG2 cells (Figure 3b), thereby suggesting that OPC can increase lipid oxidation and decrease de novo lipogenesis (DNL). It would be interesting to assess the effects of OPC treatment on hepatic fatty acid oxidation and DNL in vivo.

11) OPC treatment significantly increased the AMP/ATP ratio in ZDF rats (Figure 4c). Is pAMPK and pACC altered in this model?

12) Does OPC treatment affect hepatic glucose production in vivo? Are there differences in hepatic acetyl-CoA levels?

13) An on-target effect of mitochondrial uncouplers is increased body temperature. Did the authors note any changes in body temperature, especially at the higher doses of OPC tested?

14) The authors report that OPC treatment lowered blood pressure, delayed stroke-onset, extended survival and ameliorated albuminuria in salt-loaded SHRSPs (Figure 5), however they provide no direct mechanism for these observations. AMPK activation and ROS should be assessed in the vascular endothelium and kidney of SHRSP-treated rats. Did levels of nitric oxide change with OPC treatment?

15) The manuscript should be edited by a native English speaker.

Reviewer #2 (Remarks to the Author):

In their manuscript Kanemoto et al. describe the discovery of a new small molecule (OPC-163493) that acts as a mitochondrial uncoupler. Unlike other mitochondrial uncouplers, that can have deleterious side effects, the new compound seems safer presumably due to its restricted bioavailability. The authors show that OPC can improve diabetic symptoms in multiple rodent models for both T1 and T2 Diabetes. The authors describe the metabolomic changes induced by OPC both in-vitro and in-vivo, which generally demonstrate increased AMP/ATP ratio, glycolysis, TCA cycle anaplerosis, and respiration via complex I. to further

support a therapeutic potential the authors also show beneficial cardiovascular effects. The concept of using mitochondrial uncouplers to treat diabetes has been introduced in the past and finding safe uncouplers is indeed an interesting approach for the development of new therapies for this disease. Since the concept has been already tested with different uncouplers it is important to specifically define more in depth how this uncoupler functions both in vitro and in vivo:

1. According to the authors' hypothesis the improved metabolic phenotype is induced by increased respiration through complex I, III and IV. The metabolomics data can support this; however, it is not completely clear whether the anti-diabetic effects in-vivo are a result of this. This is especially important since the authors do not see any changes in energy expenditure and show only a small difference in OCR in-vivo (Figure 1g). The authors should at least try and test if isolated mitochondria from liver respire differently upon OPC treatment using either pyruvate (complex I substrate) or succinate (complex II) as substrates to show increased activity of complex I.
2. There is no evidence that the in vivo metabolic effects are due to hepatocyte uncoupling respiration. This is an important point as the compound might target other tissues that have not been investigated such as brown or beige fat. Also, what is the basis of an uncoupler being tissue-specific? Is the potential target liver specific?
3. It is not clear what results in the improved glycemic index. Is it increased hepatic glucose utilization? Reduced hepatic glucose production? Improved insulin sensitivity due to reduced steatosis? The authors should discuss it more clearly in the text and provide experimental evidence to support this (for instance, is insulin signaling in the liver improved? is glucose production reduced? glycogen content change etc.)
4. The authors show improvement in hepatic steatosis, probably due to increased lipid oxidation, however only in the ob/ob mice. Is hepatic steatosis also improved in the other models used in this manuscript? This is important since the prediction would be that this uncoupler would improve hepatic steatosis in all models.
5. Assessment of potential liver toxicity is lacking. Liver histology and plasma levels of liver enzymes will help to support that no liver damage is caused.
6. All the metabolomic data is interesting, but without functional analysis in vivo is of difficult interpretation.
7. The manuscript is VERY difficult to read. The rationale behind the experiments and the conclusions made, although scientifically valid, are not well explained. Substantial editing is necessary to make it accessible to a broader audience.

Reviewer #3 (Remarks to the Author):

This study examines the effects of a putative mitochondrial uncoupler (OCT-16349), the action of which is localised to the liver. Studies are performed in multiple models of diabetes, metabolic syndrome and hypertension. This is a large body of work, demonstrating that OPC-163493, reduced diabetic complications possibly by increasing hepatic glucose utilisation and suppression of glucose production. In addition, in a model of salt-sensitive hypertension OCT-163493 was shown to cause a moderate reduction in blood pressure and ameliorate albuminuria. The compound shows potential as a candidate for treatment of metabolic diseases such as diabetes.

GENERAL COMMENT:

Overall, the studies are comprehensive, well conducted and results accurately reported. The English expression is poor and therefore readability is low. In part, this due to the extensive amount of work being presented and the need to adhere to the word limits. However, these errors make it difficult to understand the findings. There are sentences that do not make any sense (I give a few examples) and grammatical errors result in confusion.

Examples of poorly written or incomprehensible sentences

In the abstract – when first read, I thought that the majority of the abstract was describing studies that had been completed prior to this study – problems with grammatical tense.

Introduction - Aiming to treat DM with a mUncoupler, focusing on the liver as a specific target organ is reasonable in a sense of safety similar to a strategy proposed by Shulman et al.

First paragraph discussion page 18: Our preclinical toxicity study revealed OPC-163493 to have remarkably higher no observable adverse effect level (NOAEL) than its effective dose and level (Supplementary Fig. 6).

Final sentence discussion Page 21-“Especially, challenging clinical trials of OPC-163493

and other safe

uncouplers could prove their usefulness and safety for humans, and revive a mechanism of mUncoupling as a therapeutic strategy for treating metabolic diseases. “

SPECIFIC COMMENTS:

1. What is the impact of uncoupling mitochondrial respiration in the kidney? Given the very high dependence of renal reabsorptive capacity on ATP production, and that the OCT163439 accumulates in the kidney like the liver, this requires greater investigation. le what was OCR in the kidney? Where tissues examined for histopathology? Impact on sodium excretion?

a. In the study in salt load SP-SHR OCT-163439 treated group food intake more than doubled, indicating greater sodium intake, without a greater urine output if anything it fell (decrease 15%- not significant) – this suggests fluid retention. This findings in light of the reduction in BP should be discussed. What is the mechanism of for BP decrease in this model? Is it driven by changes in cardiac output or total peripheral resistance?

b. In future studies the blood pressure effects of OCT163439 will need to be confirmed by radiotelemetry (gold standard).

2. Blood pressure measurement:

a. It would be helpful if effect size was indicated in the main text- I estimate that the reduction in SBP was ~10 mmHg., which is at the detection limit for the tail cuff method.

b. The authors have a previous publication in which they validate their method for measuring SBP measurement against telemetry recordings in rats.

c. Measurements via tail-cuff have on numerous occasions been reported not to accurately reflect diastolic BP and thus DBP should be omitted.

d. It is surprising that SBP was not measured in the other models- this would have strengthened the data.

e. In Figure 5b the description of the statistical analysis does not make sense. “Each time point was evaluated by comparison of group × time interaction (**P < 0.01).

Does the ** in the graph refer to the interaction term from the repeated measures ANOVA (factors group, time and interaction) or does the symbol represent post-hoc analysis at each time point? The fact that the ** appear at 2 points on the graph suggests post-hoc analysis.

In general, the statistical analysis is poorly described leaving the reader to “guess’ what comparisons, and by what test, have been made.

Minor concerns:

Figure legends- should provide all the information necessary for the reader to interpret the graphs, but should not provide an interpretation which should be provided in the main text.

Response to Reviewers' comments:

First of all, we would like to thank the reviewers sincerely for constructive criticism and valuable comments, which have helped to improve the quality of this manuscript. Accordingly, we have systematically improved the revised manuscript, with additional experiments and new information. Our responses to the reviewers' comments are given below.

Reviewer #1 (Remarks to the Author):

In this study Kanemoto et al. identified OPC-163493 as a novel liver-localized mitochondrial uncoupler and evaluated its potential use for the treatment of T2D. Specifically, they demonstrate that oral administration of OPC-163493 significantly lowered Hba1c in multiple animal models of diabetes including ZDF rats (T2D), Akita mice (T1D), OLETF rats (late-onset T2D), and ob/ob mice (severely obese T2D) independent of changes in body weight. To further characterize the mechanism for OPC-163493's anti-diabetic effect, the authors went on to perform metabolomic analysis in HepG2 cells and ZDF rats after OPC treatment. Consistent with its role as a mitochondrial uncoupler, OPC treatment increased the AMP/ATP ratio and activated AMPK. OPC treatment also increased the GSH/GSSG ratio, indicative of reduced oxidative stress. Lastly, Kanemoto et al. assessed the effects of OPC-163493 on CV function and found that OPC treatment lowered blood pressure, extended survival and ameliorated albuminuria in salt-loaded spontaneous hypertensive rats.

Comments

Overall, this is an interesting and timely manuscript that describes the identification of a liver-directed mitochondrial uncoupler, OPC-163493, and its potential use as a novel therapy for diabetes and CVD. However, the experiments do not go far enough in demonstrating the mechanism by which this mitochondrial uncoupler ameliorates diabetes and improves CV function in the multiple animal models that were tested. The manuscript would benefit greatly from a more careful characterization of the anti-diabetic effects of OPC in vivo. Specific issues are outlined below:

- 1) The authors demonstrate that OPC-163493 is a direct mitochondrial uncoupler by

measuring OCR in HepG2 cells in the presence of oligomycin (Figure 1d-e). Given the subsequent in vivo studies, does OPC increase OCR in primary rat or mouse hepatocytes?

Response to Reviewer's Comment 1)

We measured OCR in rat primary hepatocytes and show that OPC also augmented OCR in these cells (Supplementary Fig. 1c).

2) While OPC accumulates primarily in the liver (Figure S2b), a significant amount of OPC also gets into the kidney. Markers of renal toxicity (BUN and creatinine) should be assessed in the diabetic models treated with OPC.

Response to Reviewer's Comment 2)

In diabetic animal models treated with OPC, we observed no toxicological change at doses in the pharmacological range. Moreover, we demonstrated that OPC treatment improved BUN, plasma Cr, Ccr, and albuminuria in salt-loaded SHRSPs (Fig. 6d, Supplementary Figs. 7c, 7d, 7e, Supplementary Tables 5a and 5b).

It was not until we reached a dose of 300 mg/kg that we found renal toxicity (BUN elevation in males and histological change in females) in a four-week dosing toxicity study in SD rats (Supplementary Tables 6a and 6c).

3) Mitochondrial uncouplers are expected to increase lipid oxidation and reduce intracellular lipid content. However, the authors only demonstrate that hepatic triglycerides are reduced in ob/ob mice after OPC treatment (Figure 2k). Does OPC treatment reduce liver triglycerides in the other animal models of T2D, such as ZDF rats?

Response to Reviewer's Comment 3)

All DM model animals that were described in the first manuscript didn't show steatosis—apart from ob/ob mice. In our experience with ZDF rats provided by Charles River Japan, hepatic TG contents were generally less than 10 mg/g. To further pursue this issue, we conducted a new, additional experiment where we used OPC-163493 in a high-fat-diet-induced hepatic steatosis model in SD rats, and carried out lipidomic analysis of the liver (Supplementary Table 2k).

4) The authors measure plasma concentrations of OPC and use QWBA to measure relative levels of OPC in vivo (Figure S2). It would be instructive to quantitate actual levels of OPC to determine what concentration of hepatic OPC is needed to produce anti-diabetic effects in vivo.

Response to Reviewer's Comment 4)

We have added data on hepatic content of OPC, measured directly and compared with the

plasma content (Supplementary Fig. 2c and Supplementary Table 1c).

5) The effects of OPC on HbA1c and plasma glucose levels in ZDF rats and ob/ob mice (Figure S3a,s) are relatively modest compared to the effects observed in previous studies of mitochondrial uncouplers (Perry et al. Cell Metabolism 2013, Tao et al. Nature Medicine 2014, Perry et al. Science 2015, Abulizi et al. FASEB J. 2017). Moreover, OPC failed to significantly reduce plasma insulin concentrations in the models that were tested. These studies should be discussed in the context of the present study

Response to Reviewer's Comment 5)

In the case of ob/ob mice, we observed an unanticipated increase in appetite in the OPC-treated groups that presented in a dose-dependent manner; 5% increase in 0.01% chow group (not significant) and 15% increase in 0.02% group (statistically significant). As is well known, food intake directly affects glycemic control. We speculate that the hyperphagia in the high-dose group would counteract the HbA1c-lowering effect of OPC (Supplementary Fig. 3s).

Regarding ZDF rats, we think that the rats obtained from Charles River Japan showed more severe changes in condition than the Charles River US rats. For example, in Perry's report fasting plasma insulin was more than 200 $\mu\text{U}/\text{mL}$ (Perry et al. Science 2015, Fig. 2D); corresponding values in the animals we used were about 50 and 25 $\mu\text{U}/\text{mL}$ (2 and 1 ng/mL) at the ages of 17 and 33 weeks, respectively (Supplementary Figs. 3b and 3h). The insulin depletion indicated that the animals we used developed more advanced deterioration of pancreatic β -cells. As the Japanese ZDF line demonstrates advanced diabetes, the Japanese breeder established another, more mild line, ZDF(M) rats. OPC showed potent efficacy on HbA1c and blood glucose levels in ZDF(M) rats (Fig. 2f and Supplementary Fig.4).

We don't think OPC directly affects skeletal muscle insulin sensitivity due to the organ specificity (discuss more in the response to #7); therefore, OPC didn't alter plasma insulin levels in ZDF rats (Supplementary Figs. 3b and 3h).

6) Metabolic cage studies were only completed over the course of 26 h- this should be extended to 2-3 days to give animals time to acclimate to the metabolic cage system, as well as track the metabolic impact of OPC treatment during the light/dark cycles.

Response to Reviewer's Comment 6)

In a preliminary study, after four weeks of treatment, animals were accommodated in the chambers for 72 h, and we measured VO_2 and VCO_2 for the first 24 h and the last 24 h. No effects of OPC treatment were observed with either measurement. Therefore, we

adopted a 26 h measurement protocol in the final study.

7) The authors suggest that the anti-diabetic effects of OPC are due to alterations in the hepatic metabolome. Given that mitochondrial uncouplers have previously been found to improve whole-body glucose metabolism by improving liver and skeletal muscle insulin sensitivity (Perry et al. Science 2015 and Tao et al. Nature Medicine 2014), it would strengthen the manuscript if the authors performed hyperinsulinemic-euglycemic clamp studies to directly assess tissue-specific insulin responsiveness in these models in vivo.

Response to Reviewer's Comment 7)

In response to this comment, we conducted a hyperinsulinemic-euglycemic clamp test using ZDF(M) rats. Although OPC-163493 treatment significantly lowered the rate of hepatic glucose production (HGP) at the basal state, it didn't affect the rate of whole-body glucose disappearance (Rd) at the clamp state (Fig. 3 and Supplementary Table 3). Since organ specificity of OPC is so strict, we think that OPC-163493 mainly works on the liver and reduced the basal HGP, but doesn't directly improve muscle insulin resistance.

8) In Figure 2k, the authors demonstrate that OPC treatment significantly reduces hepatic triglycerides. Given the potential role of hepatic ceramides, diacylglycerols, acylcarnitines in mediating hepatic insulin resistance, the authors should measure these metabolites in liver and skeletal muscle. The authors should also assess the effects of OPC treatment on plasma adipokines/cytokines (adiponectin, leptin, resistin, IL-6, TNF- α).

Response to Reviewer's Comment 8)

As described in the response to #3, we performed lipidomic analysis in the liver of HFD SD rats. The experiment showed that OPC reduced hepatic diacylglycerols and acylcarnitines but did not alter ceramides (Fig. 2m). Likely because OPC has little effect on fat, we did not find any change in plasma adipokines levels (Supplementary 3w). Neither IL-6 nor TNF- α levels were measurable in the plasma of HFD SD rats using commercially available ELISA kits due to sensitivity limitations.

9) No differences in plasma triglycerides were observed in ob/ob mice treated with OPC (Figure 2j). Did OPC treatment alter plasma triglycerides in any of the other diabetic models? Was total plasma cholesterol or LDL-cholesterol altered?

Response to Reviewer's Comment 9)

In our study of HFD SD rats, OPC didn't affect plasma triglyceride or total cholesterol.

10) The authors demonstrate that OPC treatment significantly increases pAMPK and pACC

in HepG2 cells (Figure 3b), thereby suggesting that OPC can increase lipid oxidation and decrease de novo lipogenesis (DNL). It would be interesting to assess the effects of OPC treatment on hepatic fatty acid oxidation and DNL in vivo.

Response to Reviewer's Comment 10)

Unfortunately, we couldn't conduct an in vivo study, but elucidated that OPC increased FAO-dependent OCR in HepG2 cells (Fig. 1g and Supplementary Fig. 1d).

11) OPC treatment significantly increased the AMP/ATP ratio in ZDF rats (Figure 4c). Is pAMPK and pACC altered in this model?

Response to Reviewer's Comment 11)

Since our animal care and use committee didn't allow an additional study of ZDF rats merely to take Western blots samples, we prepared liver samples from HFD SD rats. The result is shown in below. Unfortunately, we could detect only a tendency towards an increase in pAMPK.

12) Does OPC treatment affect hepatic glucose production in vivo? Are there differences in hepatic acetyl-CoA levels?

Response to Reviewer's Comment 12)

The hyperinsulinemic-euglycemic clamp test showed that OPC reduced the basal HGP (the response to #7). We added the graph "Metabolic analysis of in vivo effects of OPC-163493 in ZDF rats" and effect on acetyl-CoA are shown in Supplementary Fig. 6d. Acetyl-CoA was significantly increased in the vehicle-treated group over six weeks (11-17 weeks old); in the OPC-treated group a tendency towards reduced levels was seen.

13) An on-target effect of mitochondrial uncouplers is increased body temperature. Did the authors note any changes in body temperature, especially at the higher doses of OPC tested?

Response to Reviewer's Comment 13)

In the toxicity studies in rats, we didn't observe an increase in body temperature until dosage reached 300 mg/kg, probably due to the organ specificity of OPC (Supplementary tables 6a and 6c).

14) The authors report that OPC treatment lowered blood pressure, delayed stroke-onset, extended survival and ameliorated albuminuria in salt-loaded SHRSPs (Figure 5), however they provide no direct mechanism for these observations. AMPK activation and ROS should be assessed in the vascular endothelium and kidney of SHRSP-treated rats. Did levels of nitric oxide change with OPC treatment?

Response to Reviewer's Comment 14)

Since Zhang et al. reported that CCCP directly relaxed SD rat aortas constricted by phenylephrine and that the vasodilation was endothelium-independent, we investigated whether OPC had a similar effect or not. While OPC directly relaxed rat aortas very little in the range in which lowered blood pressure was seen (up to 10 $\mu\text{mol/L}$), it improved NO availability and the improvement was more prominent in the presence of endothelia (Figs. 7a, 7b, Supplementary Figs. 8c and 8d).

This result, we think, indicates that OPC may reduce NO attrition caused by ROS, especially in the endothelia. However, the present study has not adequately addressed the direct mechanism. Further studies are needed in order to look into involvement of AMPK activation and ROS in this mechanism.

Zhang et al., Br J Pharmacol 173, 3145-3158 (2016).

15) The manuscript should be edited by a native English speaker.

Response to Reviewer's Comment 15)

The revised manuscript has been edited by a native English speaker.

Reviewer #2 (Remarks to the Author):

In their manuscript Kanemoto et al. describe the discovery of a new small molecule (OPC-163493) that acts as a mitochondrial uncoupler. Unlike other mitochondrial uncouplers, that can have deleterious side effects, the new compound seems safer presumably due to its restricted bioavailability. The authors show that OPC can improve diabetic symptoms in multiple rodent models for both T1 and T2 Diabetes. The authors describe the metabolomic

changes induced by OPC both in-vitro and in-vivo, which generally demonstrate increased AMP/ATP ratio, glycolysis, TCA cycle anaplerosis, and respiration via complex I. to further support a therapeutic potential the authors also show beneficial cardiovascular effects. The concept of using mitochondrial uncouplers to treat diabetes has been introduced in the past and finding safe uncouplers is indeed an interesting approach for the development of new therapies for this disease. Since the concept has been already tested with different uncouplers it is important to specifically define more in depth how this uncoupler functions both in vitro and in vivo:

1. According to the authors' hypothesis the improved metabolic phenotype is induced by increased respiration through complex I, III and IV. The metabolomics data can support this; however, it is not completely clear whether the anti-diabetic effects in-vivo are a result of this. This is especially important since the authors do not see any changes in energy expenditure and show only a small difference in OCR in-vivo (Figure 1g). The authors should at least try and test if isolated mitochondria from liver respire differently upon OPC treatment using either pyruvate (complex I substrate) or succinate (complex II) as substrates to show increased activity of complex I.

Response to Reviewer's Comment 1)

We measured OCR of mitochondria isolated from the livers of SD rats, and examined the effect of OPC treatment using pyruvate/malate or succinate as substrates. As shown in the figure below, OPC treatment augmented mitochondrial respiration under both substrate conditions. We think that increased respiration through complexes I, III and IV occurred as a cellular adaptation of HepG2 cells. OPC-163493 may incite cells to use the most efficient catabolic (respiratory) mechanisms they can and promote optimal usage of available energy sources within cells, to maintain $\Delta\psi$. As another example of the utilization of a wide variety of energy sources, OPC treatment augmented OCR by enhancing FAO in HepG2 cells which have been cultured in media depleted of glucose (Supplementary Fig. 1d and Fig. 1g).

2. There is no evidence that the in vivo metabolic effects are due to hepatocyte uncoupling respiration. This is an important point as the compound might target other tissues that have not been investigated such as brown or beige fat. Also, what is the basis of an uncoupler being tissue-specific? Is the potential target liver specific?

Response to Reviewer's Comment 2)

We conducted a hyperinsulinemic-euglycemic clamp test using ZDF(M) rats. Although OPC-163493 treatment significantly lowered the basal rate of hepatic glucose production (HGP), it didn't alter the rate of whole-body glucose disappearance (Rd) in the clamped state (Fig. 3 and Supplementary Table 3). The results suggested that OPC-163493 mainly worked on the liver and reduced the basal HGP but did not improve muscle and fat insulin resistance in ZDF(M) rats due its organ specificity.

3. It is not clear what results in the improved glycemic index. Is it increased hepatic glucose utilization? Reduced hepatic glucose production? Improved insulin sensitivity due to reduced steatosis? The authors should discuss it more clearly in the text and provide experimental evidence to support this (for instance, is insulin signaling in the liver improved? is glucose production reduced? glycogen content change etc.)

Response to Reviewer's Comment 3)

As described above, hyperinsulinemic-euglycemic clamp test showed OPC treatment significantly lowered the rate of HGP.

4. The authors show improvement in hepatic steatosis, probably due to increased lipid oxidation, however only in the ob/ob mice. Is hepatic steatosis also improved in the other models used in this manuscript? This is important since the prediction would be that this

uncoupler would improve hepatic steatosis in all models.

Response to Reviewer's Comment 4)

None of the DM model animals that were described in the first manuscript showed steatosis, apart from ob/ob mice. To examine the effect of OPC on hepatic steatosis, we carried out another animal experiment using High-fat-diet SD rats. In this experiment, lipidomic analysis in the liver showed that OPC treatment improved hepatic steatosis.

5. Assessment of potential liver toxicity is lacking. Liver histology and plasma levels of liver enzymes will help to support that no liver damage is caused.

Response to Reviewer's Comment 5)

We added a section "Safety assessment of OPC-163493." As described in that section, we didn't observe any histological change in the liver until a dose of 100 mg/kg was attained, and the change was limited to mild hypertrophy of hepatocytes. ALT and AST elevation was only found at 300 mg/kg (Supplementary Tables 6a and 6c).

6. All the metabolomic data is interesting, but without functional analysis in vivo is of difficult interpretation.

Response to Reviewer's Comment 6)

We believe that the data obtained from the hyperinsulinemic-euglycemic clamp test could help interpretation of in vivo effects.

7. The manuscript is VERY difficult to read. The rationale behind the experiments and the conclusions made, although scientifically valid, are not well explained. Substantial editing is necessary to make it accessible to a broader audience.

Response to Reviewer's Comment 7)

The manuscript has been revised, and edited by a native English speaker.

Reviewer #3 (Remarks to the Author):

This study examines the effects of a putative mitochondrial uncoupler (OCT-16349), the action of which is localised to the liver. Studies are performed in multiple models of diabetes, metabolic syndrome and hypertension. This is a large body of work, demonstrating that OPC-163493, reduced diabetic complications possibly by increasing

hepatic glucose utilisation and suppression of glucose production. In addition, in a model of salt-sensitive hypertension OCT-163493 was shown to cause a moderate reduction in blood pressure and ameliorate albuminuria. The compound shows potential as a candidate for treatment of metabolic diseases such as diabetes.

GENERAL COMMENT:

Overall, the studies are comprehensive, well conducted and results accurately reported. The English expression is poor and therefore readability is low. In part, this due to the extensive amount of work being presented and the need to adhere to the word limits. However, these errors make it difficult to understand the findings. There are sentences that do not make any sense (I give a few examples) and grammatical errors result in confusion.

Examples of poorly written or incomprehensible sentences

In the abstract – when first read, I thought that the majority of the abstract was describing studies that had been completed prior to this study – problems with grammatical tense.

Introduction - Aiming to treat DM with a mUncoupler, focusing on the liver as a specific target organ is reasonable in a sense of safety similar to a strategy proposed by Shulman et al.

First paragraph discussion page 18: Our preclinical toxicity study revealed OPC-163493 to have remarkably higher no observable adverse effect level (NOAEL) than its effective dose and level (Supplementary Fig. 6).

Final sentence discussion Page 21-“Especially, challenging clinical trials of OPC-163493 and other safe uncouplers could prove their usefulness and safety for humans, and revive a mechanism of mUncoupling as a therapeutic strategy for treating metabolic diseases. “

Response to Reviewer’s General Comment)

The revised manuscript has been edited by a native English speaker. We believe that the English has improved and that the findings contained in the manuscript are now more intelligible.

SPECIFIC COMMENTS:

1. What is the impact of uncoupling mitochondrial respiration in the kidney? Given the very high dependence of renal reabsorptive capacity on ATP production, and that the OCT163439 accumulates in the kidney like the liver, this requires greater investigation. What was OCR in the kidney? Where tissues examined for histopathology? Impact on sodium excretion?

Response to Reviewer's Comment 1)

Unfortunately, we couldn't measure renal OCR because kidney homogenates somehow lose the capacity for mitochondrial respiration very rapidly after tissue isolation. In the rat toxicity study, we didn't observe any toxicological changes in the kidney until a dose of 300 mg/kg was attained. The observed changes were BUN elevation in males, and hyperplasia of transitional cells in the renal pelvis and urinary bladder in females (Supplementary Tables 6a and 6c).

a. In the study in salt load SP-SHR OCT-163439 treated group food intake more than doubled, indicating greater sodium intake, without a greater urine output if anything it fell (decrease 15%- not significant) – this suggests fluid retention. This findings in light of the reduction in BP should be discussed. What is the mechanism of for BP decrease in this model? Is it driven by changes in cardiac output or total peripheral resistance?

Response to Reviewer's Comment 1a)

We don't think that excessive sodium intake occurred in the OPC-treated group. Since drinking water included 1% NaCl (MF chow contains 0.19 g sodium/100 g), sodium intakes of control and OPC-treated groups were calculated to be almost the same (0.61 g/24 h). In this revised manuscript, we add data related to the use of rat thoracic aortas. The experiment showed that OPC improved NO bioavailability in these aortas (Figs. 7a, 7b, Supplementary Figs. 8c and 8d). We therefore think that one explanation for OPC's BP-lowering effect is that OPC may improve access of NO to vascular smooth muscle, by reducing the attrition of NO in endothelia.

b. In future studies the blood pressure effects of OCT163439 will need to be confirmed by radiotelemetry (gold standard).

Response to Reviewer's Comment 1b)

We agree that blood pressure measurement by radiotelemetry will be necessary in future studies. Since the establishment of the techniques is thought to take a long time, we currently present further experimental results acquired using the tail cuff method to reinforce our data and the reliability of the method (Supplementary figure 7a and Supplementary table 5b).

2. Blood pressure measurement:

a. It would be helpful if effect size was indicated in the main text- I estimate that the reduction in SBP was ~10 mmHg., which is at the detection limit for the tail cuff method.

Response to Reviewer's Comment 2a)

Utilizing the SD of our experimental data (average SD = 14.06 mmHg) to determine effect sizes, we estimated respective group sizes of $n = 33$ and 15 , in order to identify differences of 10 and 15 mmHg SBP respectively (Power = 0.8, unpaired t-test). Therefore, we think a practical target to distinguish an SBP difference is 15 mmHg, using our measurement method. This estimation is described in the Methods section.

b. The authors have a previous publication in which they validate their method for measuring SBP measurement against telemetry recordings in rats.

Response to Reviewer's Comment 2b)

The previous publication was not from our group. Kubota et al. we referred to are not our colleagues.

c. Measurements via tail-cuff have on numerous occasions been reported not to accurately reflect diastolic BP and thus DBP should be omitted.

Response to Reviewer's Comment 2c)

Following the reviewer's suggestion, DBP data have been omitted.

d. It is surprising that SBP was not measured in the other models- this would have strengthened the data.

Response to Reviewer's Comment 2d)

In an additional experiment using HFD SD rats, we measured BP in normotensive rats after 11 days' treatment. We did not however find any differences between OPC- and Vehicle-treated groups.

e. In Figure 5b the description of the statistical analysis does not make sense. “Each time point was evaluated by comparison of group × time interaction (**P < 0.01).

Does the ** in the graph refer to the interaction term from the repeated measures ANOVA (factors group, time and interaction) or does the symbol represent post-hoc analysis at each time point? The fact that the ** appear at 2 points on the graph suggests post-hoc analysis.

Response to Reviewer’s Comment 2e)

We apologize for not describing things clearly in this figure legend. The ** refers to the results of a post-hoc test. We have rewritten the text as follows:

As significant decreases were found in the overall treatment differences in SBP by the MMRM method (0.06% OPC vs Control chow group, P < 0.01), treatment-by-time interaction was then estimated by contrast-averaging the corresponding treatment differences at each time point using the MMRM method (**P < 0.01).

In general, the statistical analysis is poorly described leaving the reader to “guess’ what comparisons, and by what test, have been made.

Response to Reviewer’s Comment)

We have tried to improve our descriptions as much as possible.

Minor concerns:

Figure legends- should provide all the information necessary for the reader to interpret the graphs, but should not provide an interpretation which should be provided in the main text.

Response to Reviewer's minor concerns)

These have been corrected to the best of our ability.

Reviewers' comments:

Reviewer #1 (Remarks to the Author):

The authors have done an excellent job of addressing all of my comments.

Reviewer #2 (Remarks to the Author):

For the most part the authors have addressed the previous critiques raised in the review. It will strength the manuscript to clarify:

1- Since mitochondrial uncouplers have been previously shown to improve insulin sensitivity in liver the authors need explain in the text why they think OPC-163493, in their hyperinsulinemic-euglycemic clamp study, does not improve HGP under clamp conditions.

2- pAkt levels in liver, after OPC treatment, as an indication of insulin sensitivity, should be added to the results.

Reviewer #3 (Remarks to the Author):

The authors have responded satisfactorily to my concerns and the manuscript has been significantly improved.

I apologise for raising a new issue at this stage:

Page 8: QWBA study: Supp fig 2B- Are the kidney, liver and GI tract levels in females lower than males? Will females require higher doses of the drug to be effective, leading to greater occurrence of side effects? This should be addressed in the discussion, particularly given that all the disease models were only performed in male animals.

Many drugs fail to translate to the market due to late identification of adverse effects in females.

Reviewer #2 (Remarks to the Author):

For the most part the authors have addressed the previous critiques raised in the review. It will strength the manuscript to clarify:

1- Since mitochondrial uncouplers have been previously shown to improve insulin sensitivity in liver the authors need explain in the text why they think OPC-163493, in their hyperinsulinemic-euglycemic clamp study, does not improve HGP under clamp conditions.

Response to Reviewer's Comment 1

Although we expected that OPC-163493 would significantly improve both GIR and HGP under the clamp condition, we could detect only a tendency towards an increase in GIR and a decrease in HGP in the experiment shown in the manuscript. The reason why we failed to detect a significant improvement in those parameters was the large data variation. The variation is thought to be mainly due to the wide range of disease severity in animals after the complicated experimental procedures including surgical manipulations; e.g. one of animals in OPC-treated group hardly needed glucose infusion in spite of the hyperinsulinemic state.

We added following sentences in the Discussion section, page 26-27; "OPC-163493, however, significantly lowered HGP at the basal state. Contrary to our expectation, OPC-163493-treatment showed only a tendency towards improvement in GIR and HGP under the clamp condition. The reason why significant differences were not detected in these parameters was thought to be due to the large data variation observed in GIR."

2- pAkt levels in liver, after OPC treatment, as an indication of insulin sensitivity, should be added to the results.

Response to Reviewer's Comment 2

We carried out Western blotting to examine the phosphorylation levels of Akt in the liver samples prepared from HFD SD rats which were treated with OPC or vehicle for 2 weeks. The result is shown below. Although there was a tendency of dose-dependent increase of the phosphorylation levels, we could not detect statistically significant changes by OPC-treatment. This is probably because the liver samples are collected from rats after 24 hours

fasting. Since the insulin levels of those rats were very low (Fig. 2I) due to the fasting, change of the phosphorylation level of Akt, an indication of insulin sensitivity, could not be detected clearly, even if OPC-treatment improved insulin sensitivity.

Reviewer #3 (Remarks to the Author):

The authors have responded satisfactorily to my concerns and the manuscript has been significantly improved.

I apologise for raising a new issue at this stage:

Page 8: QWBA study: Supp fig 2B- Are the kidney, liver and GI tract levels in females lower than males? Will females require higher doses of the drug to be effective, leading to greater occurrence of side effects? This should be addressed in the discussion, particularly given

that all the disease models were only performed in male animals.

Many drugs fail to translate to the market due to late identification of adverse effects in females.

Response to Reviewer's Comment

We think that QWBA study did not show remarkable gender differences in all tissues including kidney, liver and GI tract. In Supp fig 2b, which shows Kp values at 2 hours of QWBA study, it looks like there are gender differences especially in GI tract at first glance. However, comparing absolute values of radioactive concentration in male and female rats (Supp. Tables 1a and 1b), e. g. in the small intestine at other time points (8, 12, 24 h), radioactive concentrations are at the same levels in male and female (1954 vs 1909 ng eq./g at 8 h, 1285 vs 1257 ng eq./g at 12 h, 778.4 vs 699.5 ng eq./g at 24 h, respectively), suggesting no gender differences. Since the data were obtained from one animal at each time point, we think that the difference seen at 2h in the GI tract was due to individual difference. Consequently, we do not think that OPC-163493 has a different tissue distribution pattern in male and female.

REVIEWERS' COMMENTS:

Reviewer #2 (Remarks to the Author):

The authors have adequately addressed the previous concerns with their explanation and inclusion in the text.

Reviewer #3 (Remarks to the Author):

the authors have satisfactorily addressed my concerns.